# Loss of cytoplasmic actin filaments raises nuclear actin levels to drive INO80C-dependent chromosome fragmentation

Verena Hurst [1,10], Christian B. Gerhold[1,2,10], Cleo V. D. Tarashev[1], Kiran Challa[1,3], Andrew Seeber[1,4], Shota Yamazaki [5], Britta Knapp[6], Stephen B. Helliwell [6,8], Bernd Bodenmiller [7], Masahiko Harata[5], Kenji Shimada [1] & Susan M. Gasser [1,9] ✉

Loss of cytosolic actin filaments upon TORC2 inhibition triggers chromosome fragmentation in yeast, which results from altered base excision repair of Zeocin-induced lesions. To find the link between TORC2 kinase and this yeast chromosome shattering (YCS) we performed phosphoproteomics. YCS-relevant phospho-targets included plasma membrane-associated regulators of actin polymerization, such as Las17, the yeast Wiscott-Aldrich Syndrome protein. Induced degradation of Las17 was sufficient to trigger YCS in presence of Zeocin, bypassing TORC2 inhibition. In yeast, Las17 does not act directly at damage, but instead its loss, like TORC2 inhibition, raises nuclear actin levels. Nuclear actin, in complex with Arp4, forms an essential subunit of several nucleosome remodeler complexes, including INO80C, which facilitates DNA polymerase elongation. Here we show that the genetic ablation of INO80C activity leads to partial YCS resistance, suggesting that elevated levels of nuclear G-actin may stimulate INO80C to increase DNA polymerase processivity and convert single-strand lesions into double-strand breaks.

The ability of actin to form dynamic cytoplasmic filaments mediates a range of essential cellular processes, including cell movement, endocytosis, intracellular trafficking, cell division, and the maintenance of cell shape[1]. Beyond these well-characterized roles, actin contributes to the localization and activity of enzymes and transcriptional factors, including aldolase[2], the MKL1 (MAL) transcription factor[3], and may bind DNA repair factors, such as Replication Protein A (RPA), which binds single-strand (ss) DNA[4,5]. We note that actin cytoskeleton disorganization is a hallmark of cancer and that perturbed actin polymerization can contribute to oncogenic transformation[6–8].

In mammalian cells, globular (G-) actin shuttles between cytoplasmic and nuclear compartments[9], ensuring communication between the large complement of cytoplasmic filamentous (F-) actin and actin in the nucleus. The cytosolic-nuclear transport of actin depends on the dedicated transport receptors, importin-9[9] and exportin-6[10], which bind actin in complex with proteins like cofilin or profilin. These help regulate the distribution of G- and F-actin between compartments. The yeast *Saccharomyces cerevisiae* lacks the dedicated transporter exportin-6, and no dedicated actin importin has been identified; nonetheless, there is a pool of nuclear actin which is predominantly in complex with

[1]Friedrich Miescher Institute for Biomedical Research, Fabrikstrasse 24, 4056 Basel, Switzerland. [2]Bühlmann Laboratories AG, Baselstrasse 55, 4124 Schönenbuch, Switzerland. [3]Mechano-Genomic Group, Division of Biology and Chemistry, Paul-Scherrer Institute, Villigen, Switzerland. [4]Transition Bio Inc, 250 Arsenal St, Watertown 02472 MA, USA. [5]Lab. Molecular Biochemistry, Graduate School of Agricultural Science, Tohoku University, Aramaki Aza-Aoba 468-1, Aoba-ku, Sendai 980-8572, Japan. [6]Novartis Institutes for Biomedical Research, Novartis Pharma AG, Fabrikstrasse 22, 4056 Basel, Switzerland. [7]Institute of Molecular Life Sciences, University of Zürich, Winterthurerstrasse 190, 8057 Zürich, Switzerland. [8]Present address: Cellvie AG, Zurich, Switzerland. [9]Present address: University of Lausanne, Department of Fundamental Microbiology, and Agora Cancer Center, ISREC Foundation, rue du Bugnon 25A, 1005 Lausanne, Switzerland. [10]These authors contributed equally: Verena Hurst, Christian B. Gerhold. ✉e-mail: susan.gasser@fmi.ch

actin-related proteins, such as Arp4 (BAF53 in mammals). Indeed, yeast actin also harbors validated nuclear export signals, which allow actin to accumulate in the nucleus when mutated[11].

In both mammals and budding yeast, the actin-Arp4/BAF53 complex is an integral component of large, multisubunit nucleosome remodelers and histone acetyltransferases, such as TIP60 (EP400)/NuA4, INO80/INO80C, SRCAP/SWR1-C, and BRG/Swi-Snf[12–15]. In this context, the actin-containing ARP subcomplex either forms an essential bridge linking other subcomplexes within the large remodelers or provides essential nucleosome binding sites[12]. Actin has been proposed to be rate-limiting for the formation of a functional INO80C remodeler, and excess Arp4-actin complex appears to block nuclear actin polymerization in cultured mammalian cells[16].

Beyond its essential role in nucleosome remodelers, nuclear actin can form short, transient filaments under certain conditions[17,18]. For instance, transient actin filaments (F-actin) were observed in the nuclei of cultured mammalian cells as they transition from telophase to G1[19], and also upon growth factor stimulation[20], during heat shock[21], and in response to cytotoxic stress[22–24]. In Hela cells, formin-dependent nuclear F-actin was observed after the treatment of cells with the alkylating agent methylmethanesulfonate[23], and actin depletion led to an increase in 53BP1 foci[23], although why this occurred was unclear. More recently, transient nuclear actin filaments were implicated in restricting the DNA polymerase PRIMPOL from binding at stalled forks, thereby favoring that stalled forks restart through strand invasion[18].

As a word of caution, we note that the detection of nuclear actin filaments in these reports often made use of in vivo expression of a fluorescent high-affinity actin-binding domain fused to a nuclear localization signal (e.g., Lifeact-EN or Utr230-EN). These sensors increase the nuclear concentration of actin and can seed actin filament formation (reviewed in ref. 17). Nonetheless, both mutant and inhibitor studies suggest that regulators of actin polymerization promote mammalian double-strand break (DSB) clustering[25,26], or help shift DSBs from centromeric satellite sequence to the nuclear periphery[22]. Indeed, the metazoan actin-binding proteins ARP2 and ARP3, as well as the Wiscott-Aldrich Syndrome Protein, WASP, were recovered by Chromatin immunoprecipitation (ChIP) at DSBs, where their activities appeared to promote resection[25], clustering and inappropriate translocations[26].

Although subnuclear DSB mobility is influenced by mutations that affect actin filament formation in flies[22], there is no evidence that nuclear actin filaments are involved in the local chromatin movement that occurs in response to targeted DSB in yeast[27]. Increased DSB movement in yeast depends on activation of the checkpoint kinase Mec1/ATR[28] and the actin-containing nucleosome remodeler INO80C[29], and is constrained by cohesin[29], such that even the enhanced movement triggered by DNA damage retains characteristics of a constrained random walk[30], rather than the directional movement that one expects for movement along actin filaments. Intriguingly, in yeast, Latrunculin A (LatA)-induced F-actin depolymerization attenuated a nuclear rocking movement[30], which stems from the interaction of cytoplasmic actin filaments with the nuclear envelope-spanning LINC complex[31,32], independent of DNA damage. It remains unclear to what extent apparent differences in DSB movement in yeast and mammals reflect differences in modes of measurement (reviewed in ref. 27,33).

Earlier work from our laboratory has shown that TORC2 kinase inhibition by an imidazoquinoline (NVP-BHS345) renders yeast hypersensitive to the radiomimetic drug Zeocin, generating irreversible and randomly distributed DSBs[34]. The combinatorial treatment leads to rapid yeast chromosome fragmentation, called yeast chromosome shattering (YCS), without passage through the cell cycle. The fragmentation is independent of the two main DSB repair pathways, homologous recombination (HR) and Non-homologous end-joining (NHEJ), and is not a result of apoptosis[34]. Instead, YCS requires glycosylases and endonucleases of the base excision repair (BER) pathway and appears to stem from misregulated long-patch repair of clustered

oxidative lesions[35]. The synergistic effect of Zeocin and TORC2 inhibition is mimicked by combining Zeocin with the loss of two TORC2 effector kinases, Ypk1/Ypk2[35], or by adding LatA[35], both of which trigger cytoplasmic actin filament depolymerization.

Here, we explored in depth the proposed mechanistic link between actin depolymerization and the repair of Zeocin-induced DNA damage. We introduce a more potent TORC2 inhibitor, CMB4563 (hereafter CMB), and examine genetic mutations downstream of TORC2 that support or block YCS. Besides the loss of TORC2 effector kinases, Ypk1 and Ypk2[34,36,37], the loss of either cytoplasmic actin polymerization cofactors, yeast Pan1 or Las17 (human EPS15-like and WASP, respectively)[38,39], triggers YCS on Zeocin bypassing TORC2 inhibition. Importantly, we find that the ablation of Las17, like TORC2 inhibition, increases nuclear actin levels in yeast. Because nuclear actin is predominantly found in nucleosome-modifying complexes, we tested the role of actin-dependent nucleosome remodelers in YCS. We found that the loss of INO80C activity, a remodeler implicated in replication polymerase processivity[40–45] attenuates YCS. Based on this and data from Shimada et al.[35], we suggest that nuclear G-actin upregulates INO80C activity, thereby misregulating long-patch BER (LP-BER), which in turn converts clustered oxidative lesions into DSBs. This establishes an important role for the cellular F-/G-actin ratios and actin-dependent chromatin remodelers in genome stability.

## Results

### TORC2 inhibition triggers YCS after low-dosage Zeocin incubation

A chemicogenetic screen in yeast showed that TORC1 and TORC2 inhibition by NVP-BHS345 (hereafter BHS, Fig. 1a) was synthetic lethal in the presence of low levels of the radiomimetic chemical Zeocin or γ-irradiation[34]. This TOR inhibitor does not generate abasic sites, or ss breaks on its own, nor does it augment the level of oxidized base damage generated by a fixed concentration of Zeocin[35]. Instead, TORC2 inhibition leads to the conversion of oxidized bases into irreparable DSBs, triggering anywhere from 20 to 120 genomic DSB after a 1 h exposure of 50 to 80 μg/ml Zeocin[34]. These DSBs are monitored by neutral (non-denaturing) pulse-field gel electrophoresis (PFGE), which allows us to follow the conversion of full-length chromosome bands into chromosomal fragments of 50–200 kb in length (Fig. 1b). Here we identified a BHS-related imidazoquinoline called CMB4563 (CMB, Fig. 1a), that is roughly ten-fold more potent than BHS in the YCS assay[35]. We use the two inhibitors interchangeably, albeit at different concentrations.

To estimate the frequency of DSBs induced under our standard conditions, we modeled the number of random breaks necessary to reduce all yeast chromosomes (which range in size from 0.23 to 2 Mb) to fragments ranging from 50 to 300 kb, with an average length size around $100 \pm 20$ kbp[35]. This average fragment size theoretically requires between 80 and 140 randomly distributed DSBs[35]. It is estimated that Zeocin induces ss lesions in a roughly 10:1 ratio[46], thus the amount of Zeocin used could induce as many as 1400 single base lesions, a level that is normally repaired rapidly in yeast by base- and nucleotide-excision repair pathways. To be able to compare rates of DSB formation under varying experimental conditions, we scanned the intensity of fluorescently stained dsDNA after PFGE and divided the signal into intensities above 570 kb (A) and below 560 kb (B, see Methods). From this, we calculate a B/A ratio that robustly quantifies the relative integrity of the yeast genome: an intact genome yields a B/A ratio <0.5, while B/A values ranging from 3 to 50 can be scored after fragmentation (Methods[35]; for gel quantitation and B/A ratio calculation see Supplementary Data 1).

Haploinsufficiency profiling[47] showed that CMB, like BHS, impairs the growth of diploid yeast lacking one copy of various TORC1 and TORC2 subunits, and especially loss of the Tor2 kinase subunit[34,35]. Here we confirm that Tor2 kinase is the YCS-relevant target of CMB by challenging yeast bearing a point mutation in its conserved

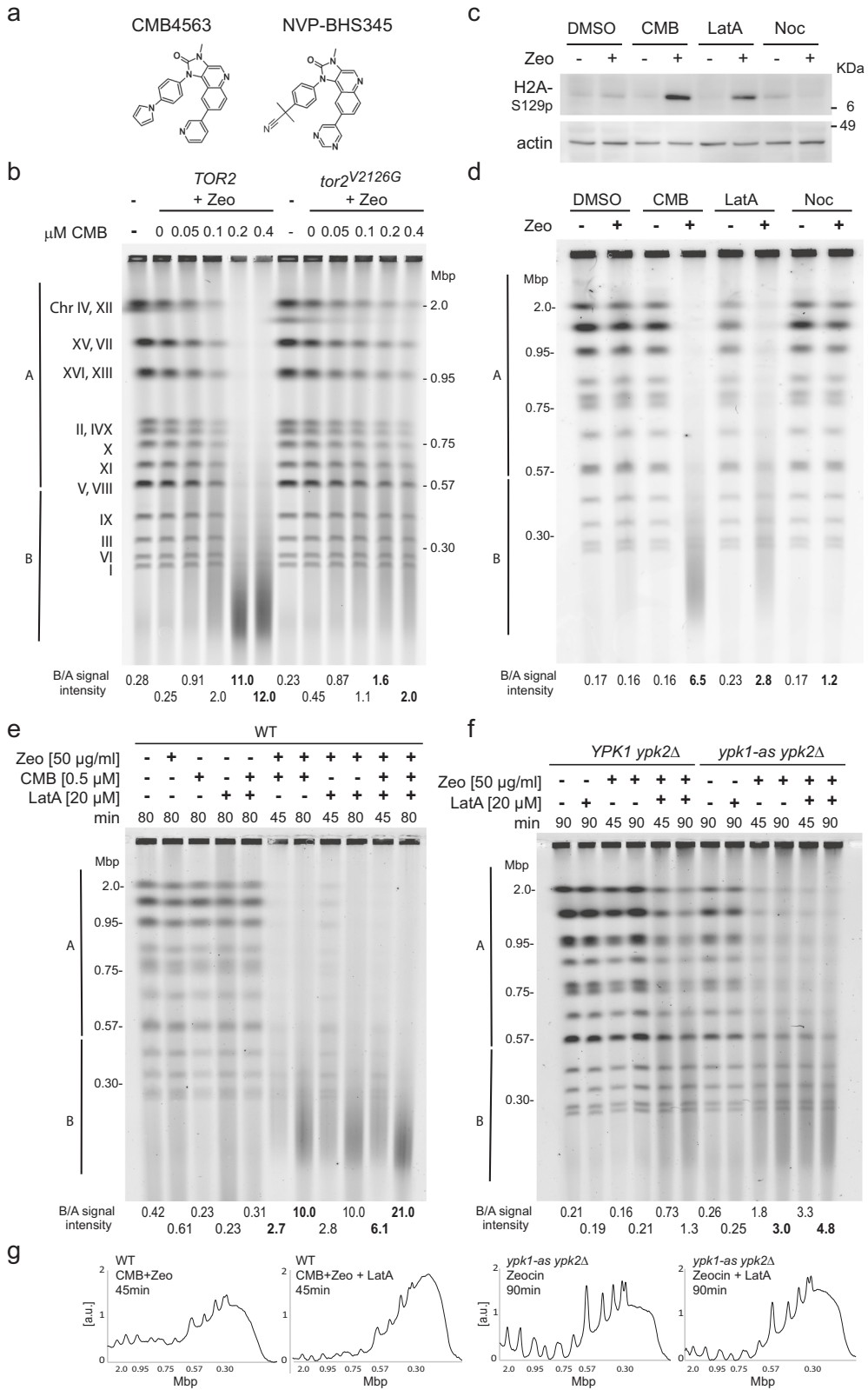

imidazoquinoline binding pocket[34], *tor2-V2126G*, with Zeocin and CMB (Fig. 1b). Yeast that express only the tor2-V2126G allele were largely resistant to CMB-induced YCS on Zeocin (Fig. 1b; B/A value = 1.6 to 2.0 vs 11 to 12 in *TOR2*+ cells). The specificity of CMB, like BHS, is underscored by showing that the related yeast PI3 kinases Mec1 and Tel1 (ATR and ATM homologs) remained functional on CMB and Zeocin, as monitored by phosphorylation of H2A-S129 (Fig. 1c). Importantly, neither low-dose Zeocin (50 μg/ml), nor up to 10 μM of CMB, led to

DSB accumulation on their own, confirming that TORC2 inhibition and Zeocin act synergistically[34].

**Ypk1/Ypk2 kinase inhibition and LatA trigger YCS on Zeocin**

The conversion of Zeocin-induced lesions into DSBs was tested with a number of other inhibitors besides CMB. A similar but less potent fragmentation was observed with the actin polymerization inhibitor LatA, but not with nocodazole, which impairs microtubule

**Fig. 1 | TORC2 inhibition and Latrunculin A act on the same pathway for Zeocin-induced YCS. a** Structures of related imidazoquinoline TORC inhibitors CMB4563(CMB) and NVP-BHS345 (BHS). **b** Isogenic GA-6148 (*TOR2*) and GA-6150 (*tor2$^{V2126G}$*) strains were treated with 75 µg/ml Zeocin (Zeo) with the indicated concentrations of CMB for 75 min. Genomic DNA was subjected to CHEF gel analysis and stained by EtBr, and quantitation is as described in Methods. The *tor2$^{V2126G}$* mutation blocks the imidazoquinoline binding pocket rendering GA-6150 resistant to CMB. Molecular weight markers and yeast chromosome numbers are indicated. B/A signal intensity is used to compare chromosome fragmentation within one gel. In bold are values discussed in the text. **c** Exponentially growing GA-1981 (WT) yeast cells were treated with either 1% DMSO, 0.5 µM CMB, 20 µM Latrunculin A (LatA), or 50 µM Nocodazole (Noc), with or without 50 µg/ml Zeocin for 80 min. Total protein extract was subjected to Western blot probed with anti-H2A-S129-phospho antibody[95] and anti-actin antibody (My BioSource, BSS9231831) or anti-tubulin antibody (Abcam, ab6161). Histone H2A-S129 is the target of checkpoint kinases Tel1 and Mec1. **d** Yeast genomic DNA from WT cells treated as in **c**) was analyzed by CHEF gel electrophoresis as in **b**) and was stained with HDGreen. Gel conditions and quantitation as **b**. **e** Exponentially growing WT cells (GA-1981) were treated with Zeocin (50 µg/ml), CMB (0.5 µM), LatA (20 µM), or with indicated combinations for 45 or 80 min. Genomic DNA was subjected to CHEF gel analysis and quantitation as in **b**. In bold are relevant lanes discussed in the text. **f** Exponentially growing *YPK1 ypk2Δ* (GA-5892) and *ypk1-as ypk2Δ* (GA-5893) cells were treated with Zeocin (50 µg/ml), LatA (20 µM), or the combination, all in the presence of 0.5 µM 1NM-PP1, which inactivates the analog sensitive (as) allele of Ypk1. Incubation time is as indicated. Genomic DNA was analyzed and quantified as in **b**. **g** The scans of signal intensities of relevant lanes (indicated in bold) are plotted for visualization but are not those used for quantitation. To the left, the scans relate lanes with bold B/A values in **e** and to the right, **f**).

polymerization (Fig. 1d, e). Earlier work showed that the inhibition of RNA pol II did not provoke YCS[34], whereas the depletion of Ypk1/Ypk2 kinases did (*ypk2Δ ypk1-as* plus 1NM-PP1, Fig. 1f)[34], although again less efficiently than TORC2 inhibition. To see if TORC2 and Ypk1/Ypk2 act on the same pathway as LatA, we checked the impact of combining inhibitors. On a fixed concentration of Zeocin, CMB showed partial additivity with LatA (B/A increases from 2.7 to 6.1 for CMB vs. CMB + LatA at 45 min; Fig. 1e, and gel traces, Fig. 1g), while Ypk1/Ypk2 ablation with LatA were less additive (B/A = 3.0 vs 4.8, for *ypk$^-$* vs *ypk$^-$* + LatA at 90 min; Fig. 1f, g). This partial epistasis suggests that the inhibition of TOR complexes by CMB does more than interfere with actin polymerization, yet all three triggers of YCS (*ypk$^-$*, LatA, and TOR inhibition) do have a negative impact on actin filaments.

## Synergy between actin depolymerization and Zeocin in human cells

Given that both TORC2 complex function[48] and BER pathways are conserved from yeast to man[49], we asked whether the synergy between Zeocin and either TOR kinase inhibition or actin filament disassembly in yeast is observed in human cells. We treated diploid primary Normal Human Dermal Fibroblasts (HDFn) for 1 h either with Zeocin alone, Latrunculin B (LatB) alone, or the two sequentially and monitored damage-induction through the intensity of H2AX phosphorylation (γH2AX)[50]. Note that LatB acts in a manner similar to LatA[51], but is more efficient than LatA in human cells. Zeocin alone produced a low γH2AX signal indicating a DNA damage response, which was enhanced by the combination of Zeocin and LatB (Fig. 2a). To see whether a combined incubation triggers cell death, as it does in yeast, we exposed HDFn cells for one hour to titrations of Zeocin, LatB, or the two combined, and monitored cell killing by Trypan blue staining and a knockoff assay (see Methods). Neither compound alone triggered cell death after 1 h, even at the highest concentration tested, whereas Zeocin combined with 300 nM LatB left ~12% of the cells dead or dying (Fig. 2b). Thus, we detected a synergistic effect of actin filament disassembly and Zeocin-induced oxidation in primary human fibroblasts. The effect is less pronounced than in yeast, which shows up to 1000-fold higher lethality when Zeocin is combined with TOR inhibition[34,35].

We confirmed by visual inspection that the treatments had the expected impact on both DNA integrity and on filamentous actin, and quantified the γH2AX damage signal and SiR-actin structure (Fig. 2c). Exposure to either 300 nM LatB, 60 µg/ml Zeocin or both for 1 h, yielded low levels of γH2AX signal on LatB or Zeocin, which increased by 50% after the combined treatment (average values are 12.3 vs 8.1 γH2AX signal in arbitrary units; Fig. 2c). The actin cytoskeleton showed the expected disassembly in the presence of LatB, and under none of these conditions did we detect a nuclear F-actin signal. The additive effects of Zeocin and LatB on the DNA damage checkpoint kinase (γH2AX readout, Fig. 2c) is consistent with an increase of both ss nicks and DSBs, as measured by the DNA tail in an alkaline Comet assay (Fig. 2d; >50% increase in Zeocin + LatB conditions).

Given that changes in the actin cytoskeleton help drive oncogenesis[6–8], we next tested whether actin depolymerization shows more synergy with Zeocin in a human cancer cell line, as opposed to the normal diploid fibroblasts used above. We chose a well-characterized human colorectal cancer cell line, HCT116, and tested serial dilutions of Zeocin vs either AZD8055, an mTOR catalytic inhibitor[52], cytochalasin D, which inhibits actin polymerization, or nocodazole, each used at sublethal concentrations[53]. We did not use the CMB or BHS compounds because in mammalian cells, unlike yeast, both inhibitors target a broad range of PIKK, and not only TOR. In this synergy monitoring assay, there was a clear drop in cell proliferation when Zeocin was combined with the TORC1/2 kinase inhibitor (AZD8055) or cytochalasin D, but not with nocodazole (Fig. 2e; see synergy values from cross titrations in Fig. 2f). We hypothesize that the loss of the mismatch repair protein hMLH1 may enhance Zeocin sensitivity in HCT116 cells[54] over that observed in HDFn cells, where repair pathway redundancy and an efficient short-patch BER pathway mediated by PARP, XRCC1 and Ligase III likely attenuate the synergy. We note that the SP-BER efficiency factors (PARP, XRCC1, and Lig III) are absent in yeast[49,55]. In conclusion, the combination of Zeocin and actin depolymerization produced a small synthetic increase in DNA damage and cell death in HDFn cells, and a pronounced synergy in cancer-derived HCT116 cells, although we do not observe the dramatic fragmentation of the genome in human cells as we do in yeast. Elsewhere we have shown that the destabilization of actin filaments by LatB or cytochalasin B in either HeLa or U2OS cells decreases SP-BER factor recruitment (XRCC1) to laser-induced damage, while inhibition of tubulin polymerization by nocodazole increased it[56], which intriguingly correlates with the changes in lethality observed in HCT116 (Fig. 2e, f).

## Cytoskeletal control proteins show altered phosphorylation during YCS

To understand what links the TORC2 pathway to DNA repair, we performed phosphoproteomics on yeast cells that either do or do not have Ypk1/Ypk2 activity, in the presence and absence of Zeocin, looking for changes that correlate with YCS (Fig. 3a). It was thought that identifying altered phosphorylation targets might shed light on the activities relevant for YCS, especially given the broad range of activities under Ypk1/Ypk2 control (actin dynamics, sphingolipid synthesis, and phosphotidylcholine turnover). Treatments were carried out as biological triplicates, and phosphotargets were quantified by a label-free method[57,58]. Among a total of 3775 detected phosphopeptides (listed in Supplementary Data 2), we considered those with ≥2-fold change over the DMSO controls and categorized these as responding either to low Zeocin alone, to the loss of YPK kinases alone, or to the combination (YCS conditions; Fig. 3b). To avoid phosphorylation events characteristic of the DNA damage checkpoint alone, we also scored the phosphoproteome in cells exposed to high levels of Zeocin (750 µg/ml), and compared these with YCS-specific hits.

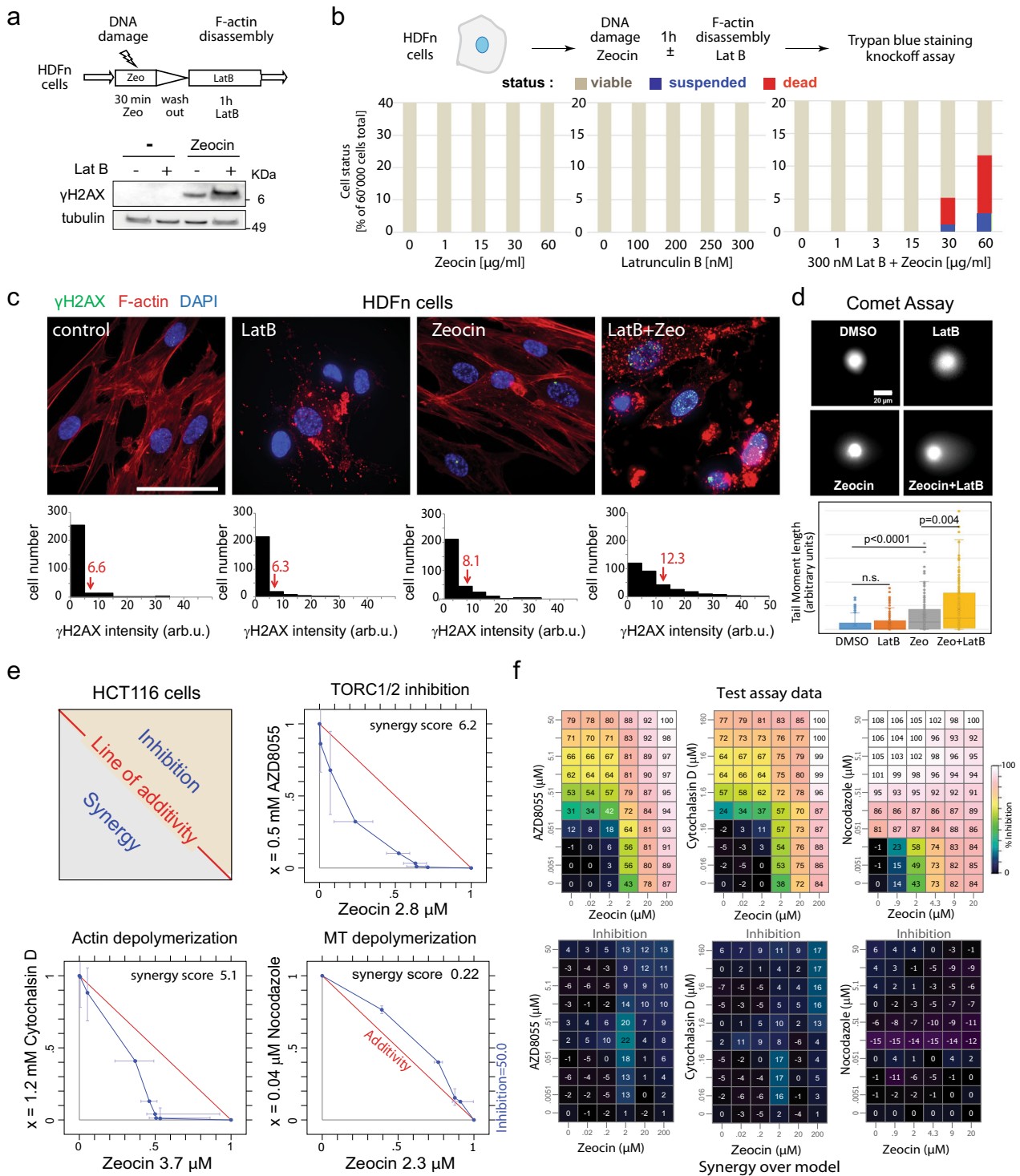

YCS-associated changes accounted for 656 up- and 305 downregulated phosphosites (YPK$^{off}$ + Zeocin; Fig. 3b), while blocking YPK activity or adding high Zeocin affected fewer phosphosites (546 up/152 down; and 322 up/103 down, respectively; Fig. 3b). The changes characteristic of YCS identified 302 proteins containing the 389 altered phosphopeptides (Fig. 3b). Not surprisingly, phosphorylation sites on low-dose Zeocin overlapped largely with those detected at high dose Zeocin (Supplemental Data 2). The GO-term most significantly represented among the 302 YCS-specific phosphorylation targets was cytoskeleton organization ($p = 2.1 \times 10^{-11}$; Fig. 3c). Although no previous study had tested YCS-specific conditions, the changes we saw upon YPK inhibition are consistent with earlier work[59] and corroborated a previous

study that used BHS345 to identify regulators of actin organization under DNA nondamaging conditions[37] (Supplementary Table 1).

Among the YCS-specific cytoskeleton-related phosphotargets we found that changes were concentrated in a set of proteins that control endocytosis and cell membrane growth through actin-dependent endo- and exocytosis[36,60] (Fig. 3d, e; Supplementary Table 1). The targets included Pan1 (EPS15-like), Sla1, Las17 (WASP), Abp1, Arp2, Sla2 and the yeast Epsins, Ent1 and Ent2, as well as two subunits of a regulatory kinase, Prk1 and Ark1 (Fig. 3e). Las17 is critical for the assembly of actin cortical patches[61], and a complex of Pan1/End3/Sla1 co-localizes with Las17 in actin patches to regulate Arp2/Arp3-dependent actin polymerization in branched structures[38,62]. All of these were

**Fig. 2 | Actin depolymerization is synthetically lethal with Zeocin in cultured human cells. a** Experimental flow. **b** Primary human dermal fibroblasts (HDFn; Life Technologies) were incubated 1 h ± 50 μg/ml Zeocin, washed, then cultured 1 h ± 300 nM LatB. Total protein extracts were probed for α-γH2AX (Merck-Milipore JBW301) and α-tubulin (Abcam ab6161) by western blot. Full blots of duplicate experiments are in Source data. **b** Synergistic lethality from Zeocin and LatB. HDFn cells (60,000 per sample) were incubated 1 h with indicated titrations of Zeocin, LatB, or in Zeocin +300 nM LatB. Dead cells (Trypan blue+) and suspended cells (Trypan blue-) were scored by BIO-TC20 (BioRad). **c** HDFn cultures were supplemented with SiR-actin (2 μM) to visualize actin, and treated 1 h with LatB (300 nM), Zeocin (60 μg/ml), or both. Fixed cells immunostained for γH2AX (green, see **a**), F-actin (SiR-actin, red), and DAPI (blue) were imaged and γH2AX intensity (arbitrary units) was measured by ImageJ for control cells (n = 294), LatB (n = 252), Zeocin (n = 299), and both (n = 323 cells). Arrows indicate mean intensities. Bar = 25 μm. **d** Alkaline Comet assay monitors ss and ds breaks in HDFn cells after 4 h on LatB (300 nM), Zeocin (60 μg/ml), or both. Comet tail moments calculated by CASP software[98] for ≥200 cells each condition and plotted. Significance was determined by Mann–Whitney rank sum test. Data presented are mean value ±SEM, cells outside 95% confidence. ns = p > 0.05. Raw images uploaded as Source data file. **e** Inhibition of mTORC1/2 or actin depolymerization reduces HCT116 cell growth synergistically with Zeocin. Human HCT116 cells seeded at 2700 cells/well in 96-well format were treated with titration of Zeocin (x axis) and AZD8055 (mTOR catalytic inhibitor), Cytochalasin D (blocks actin polymerization), or Nocodazole (depolymerizes microtubules; y axes). Proliferation was measured after 96 h using cell titer blue (compound exposure 72 h). Scheme illustrates the isobologram with the red line showing additivity, synergism in lower gray triangle, antagonism in tan, as calculated by a combined chemical genetics method[53]. Blue line = 50% inhibition as **f. f** The corresponding average values of growth inhibition (upper panels) and synergy over model for three replicates. Colors = % inhibition; for synergy calculation see ref. [53].

identified as undergoing YCS-selective changes in phosphorylation. Sla1, Las17, actin and clathrin create a complex called SLAC[63], that is inhibited by the Prk1/Ark1 and Akl1 kinases[36,64] which also showed YCS-associated changes (Fig. 3d). Among these targets, Las17 is recognized as an actin chaperone that can bind G-actin with multiple domains (WH1, WH2 and a polyproline-arginine enriched domain)[39,65]. Interaction of the Las17 WH2 domain with actin is blocked by phosphorylation on S554, which reduces F-actin at endocytic sites[66], and the dephosphorylated form of Las17 contributes to the Arp2/3-mediated formation of cortical actin patches[61].

Intriguingly, the most abundant phosphopeptide detected in Las17 both under YCS conditions and high Zeocin alone contained T380, and not the previously characterized phosphoacceptor S554[66]. The T380 residue is located between two arginine-proline stretches that are thought to allow Las17 to bind and buffer G-actin[39,66] although it is unknown whether the T380-containing linker region controls actin binding. In conclusion, YCS conditions induced phosphorylation state changes that inactivated a set of factors whose physiological role is to stimulate endocytosis through actin filament growth at the plasma membrane[60,64], yet it remained unclear how cortical actin regulation might interfere with BER to generate DSBs.

## Depletion of Pan1 or Las17, but not of related phosphotargets, induces YCS

Given evidence in mammalian cells that WASP (the Las17 homolog) and ARP2/ARP3 moonlight in DSB repair[18,22,25] and that WASP seems to regulate the RPA-ssDNA filament formation at stalled replication forks[5], we decided to test whether Las17, or one of its cofactors, regulates BER in yeast. We note that yeast Pan1 has been reported to shift to the yeast nucleus in yeast under conditions of oleate toxicity[67], while Las17 has not been reported in the yeast nuclear compartment and lacks the nuclear localization signal found in WASP[68]. We tested as well three other non-essential components of the cortical actin regulatory pathway, Sla1, Abp1, and the formin homolog, Bmi1, for roles in YCS. Our hypothesis was that the loss of one of these proteins might mimic TORC2 inhibition or possibly be resistant to it during BER-induced chromosome fragmentation.

Because both *LAS17* and *PAN1* are essential for yeast viability, we created strains with degron-tagged but functional genomic versions of *LAS17* and *PAN1*. The Las17-AID and Pan1-AID fusion proteins, the only forms expressed in the resulting strains, were rapidly degraded upon addition of the plant auxin IAA (Supplementary Fig. 1a)[69], yet neither the depletion of Las17 nor of Pan1 triggered DNA damage or genomic instability on its own (Fig. 4a, b, lanes +IAA, −Zeo; Supplementary Fig. 1b, +IAA), suggesting that neither factor is essential for BER or DSB repair. However, when combined with low concentrations of Zeocin, the loss of Las17 triggered massive chromosome fragmentation, similar to TORC2 inhibition (Fig. 4a; lanes +IAA, +Zeo). The loss of Pan1 also promoted YCS, albeit less efficiently (compare Fig. 4a, b; B/A = 16

for Las17 ablation, vs 1.9 for Pan1; Supplementary Fig. 1b, +IAA, −LatA), although we note that Las17-AID was more efficiently degraded than Pan1 (Supplementary Fig. 1a). Whereas Pan1 loss was additive with BHS (Fig. 4A; B/A = 1.9 increased to 17), loss of Las17 only increased slightly on BHS, as it was already highly efficient (Fig. 4b). The fact that Las17 and Pan1 ablation mimic TORC2 inhibition on Zeocin, argues that these may be the YCS-relevant targets of the TORC2-Ypk pathway, somehow altering BER regulation[35].

We deleted other genes involved in actin dynamics, namely those encoding Sla1, Abp1, or Bni1 (formin), which are all targets of Akl1/Ark1/Prk1 kinases[64]. None provoked strong YCS on low-level Zeocin (50 μg/ml; Supplementary Fig. 1c, d). The impact of *sla1* deletion was reminiscent of degradation of its binding partner Pan1 in that chromosome fragmentation was increased, yet it only had a pronounced effect at high Zeocin concentrations (200 μg/ml; Supplementary Fig. 1d). In contrast, the ablation of Abp1, a protein that specifically activates Arp2/Arp3, showed slight resistance to fragmentation in the absence BHS (Supplementary Fig. 1d), and the loss of formin paralleled wild-type at all concentrations of Zeocin tested (cf. WT vs *bni∆*, Supplementary Fig. 1d). In conclusion, among various F-actin regulatory factors targeted by Ypk1/Ypk2, only the loss of Las17, and to a lesser extent Pan1, bypassed TORC2 activity and triggered YCS on low-level Zeocin.

We next asked if Las17 or Pan1 degradation were on the same pathway as LatA, by monitoring fragmentation in the absence of Las17 or Pan1 with or without LatA. Whereas the shattering triggered by partial loss of Pan1 was additive with LatA (B/A, 2.7 vs 11 and 3.6 vs 25, Supplementary Fig. 1b; +IAA, +LatA), Las17 degradation and LatA were essentially epistatic, particularly at 90 min (B/A = 9.0 vs 9.8; Fig. 4c). Given Las17's role as a G-actin chaperone[39,66], this result suggests that altered actin pools, or some other common function of LatA and Las17, drives YCS.

Further refinement of this hypothesis came from the analysis of a genetic suppressor of *las17∆*. It is generally accepted that *las17∆* is lethal in yeast, yet the Ayscough laboratory reported a viable *las17∆* strain (KAY473[39]). Assuming that this strain potentially carried a suppressor mutation, we analyzed KAY 473 by genetic crossing, tetrad analysis, and genome sequencing. We confirmed the presence of an inactivating mutation in a gene called *CAP2* that suppresses the lethality of *las17∆* (Supplementary Fig. 1e). Interestingly, Cap2 is a non-essential factor that antagonizes actin polymerization[70]. To see if deletion of *cap2* suppresses YCS upon the loss of *las17*, we created a full-length *cap2* deletion, coupled it with *las17∆*, and tested the double mutant for chromosome shattering provoked by low-level Zeocin, both with and without TORC2 inhibition (+CMB). The impact of *cap2∆* alone was minor (Fig. 4d, B/A = 1.8 vs 2.2 upon TORC2 inhibition), but *cap2∆* showed a consistent and stronger suppression of YCS triggered by loss of Las17 on Zeocin (Fig. 4d, B/A = 1.6 vs. 3.5 − CMB, and 3.7 vs 6.1 + CMB). Together with the observed epistasis of Las17 degradation with LatA, the suppression *las17∆* induced YCS by *cap2∆* suggests that

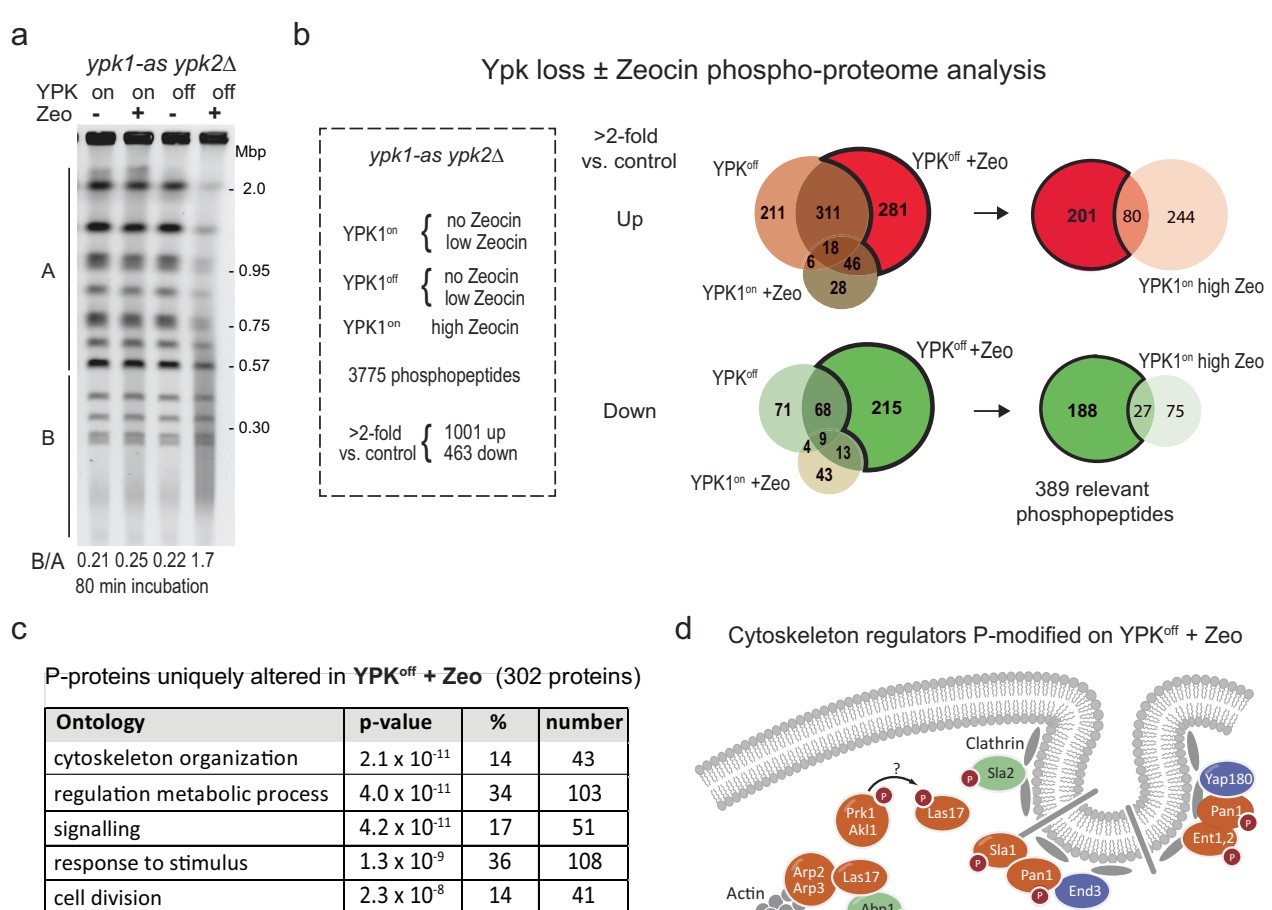

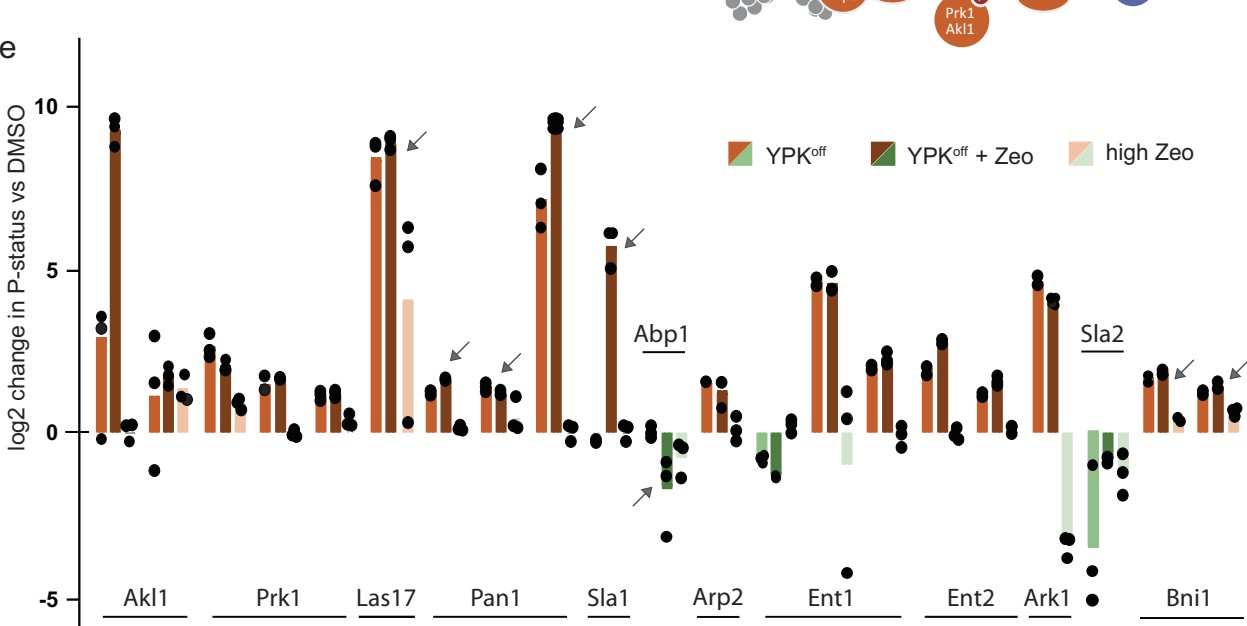

the function relevant for preventing YCS on Zeocin is related to Las17's role in actin polymerization[39].

## Las17 localization during the DNA damage response and YCS

The pathway by which Las17 regulates BER could either be direct (regulating actin at the site of damage) or indirect (by regulating global F-/G-actin balance primarily or exclusively in the cytoplasm). As

mentioned, yeast Las17 lacks the NLS that is found in mammalian WASP, which presumably promotes WASP's relocation to the nucleus under DNA damaging conditions[5,18,22,25,26]. It was, therefore, important to determine the subcellular localization of yeast Las17, particularly under DNA-damaging conditions.

We replaced the endogenous genes for *LAS17* and *PAN1*, respectively, with genomic *LAS17-GFP* and *PAN1-GFP* fusions, and monitored

**Fig. 3 | The YCS phosphoproteome identifies regulators of actin cytoskeleton as major targets. a** Exponentially growing *ypk1-as ypk2Δ* cells (GA-5893) were treated with 1% DMSO (mock), 75 µg/ml Zeocin (Ypk1^on+Zeo), 0.5 µM 1NM-PP1 (Ypk1^off), or both for 80 min, and genomic DNA was subjected to CHEF gel analysis as in Fig. 1b. **b** Treatments as in **a**, as well as wild-type yeast in 750 µg/ml Zeocin (high Zeo), were for 80 min, performed in triplicate. Extracted proteins were analyzed by label-free mass spectrometry (see Methods[58]). Phosphopeptides were triaged for change >2-fold vs DMSO control. Venn diagrams show phopshopeptides up (reddish) or down (green tones) in Ypk1^off + Zeo vs. Ypk1^off or Ypk1^on + Zeo. Ypk1^off + Zeo-specific targets are darker in the Venn diagrams. From this subgroup we subtracted phosphopeptides altered by Zeocin alone (80 up; 27 down), leaving 201 up- and 188 downregulated phosphopeptides, in 302 proteins. **c** GO-term analysis of the Ypk+Zeocin-specific phosphoproteome was carried out, and significance was determined vs a total number of factors in yeast in that category using the hypergeometric distribution. Cytoskeletal organization was most significantly enriched. **d** A cluster of interacting regulators of yeast endocytosis and actin cytoskeleton (image based on[64]), of which all except two (blue) were recovered as Ypk+Zeocin-sensitive phosphoacceptors. Red-labeled proteins gain phosphorylation, green lose it. **e** Each bar represents a specific phosphotarget site involved in actin filament regulation and endo- and exocytosis, plotted as log2 change of increased (reddish) or decreased (green) phosphorylation by indicated treatments vs DMSO. Individual values are from biological triplicates; tops of bars are median values. Relevant target sites are: Akl1: S541, S496, S504; Prk1: S553, S556, S560; Las17: T380; Pan1: S1003; S1253 or T1256; S991. T993, T995; Sla1, S449; Abp1, T206; T211; Ent1, T160, T163; T395, T388; Ent2, T468, T470, T479; Ark1. S478; Sla2, T468. T470, T479; Bni1, S325, S327, some defining overlapping phosphopeptides. Arrows indicate proteins studied further.

their localization with high resolution deconvolved fluorescence microscopy, tracking actin filament dynamics with rhodamine-coupled phalloidin. Both supported normal growth, although *LAS17-GFP* was slightly more sensitivity to Zeocin (Supplementary Fig. 1f).

In yeast, two distinct actin structures can be visualized by rhodamine-phalloidin staining. The most dominant are cytoplasmic actin cables, which are bundles of relatively long filaments that contribute to vesicle movement and cell polarity. The second is cortical actin patches, plasma membrane-associated clusters of endocytic proteins that contain short filaments, actin-binding factors, and regulatory proteins (Fig. 5a–c). In untreated cells, cytoplasmic actin cables were clearly marked by phalloidin (Fig. 5a, red in DMSO, Zeo panels), while Las17-GFP labeled the filaments weakly and strongly lit up the focal patches of actin at the plasma membrane and in emerging buds (Fig. 5a, green). Pan1-GFP is also found in the cortical actin patches and weakly labeled filaments (Fig. 5b). Under the conditions tested, the nuclei (marked by DAPI) were devoid of F-actin (phalloidin) and GFP (both Las17 and Pan1).

Upon TORC2 inhibition, Las17-GFP showed a striking relocation to small, bright round foci that are DAPI-positive, and are adjacent to, yet clearly distinct from, the cell nucleus (Fig. 5a, panels CMB). Mito-Tracker dye and quantitation of the correlation coefficient showed that these structures are mitochondria (Fig. 5d, e). On Zeocin alone, on the other hand, Las17 and Pan1 remained strongly associated with cortical actin foci and filaments near the plasma membrane (Fig. 5a, b). Under YCS conditions (CMB + Zeo), not only are cytoplasmic actin filaments lost presumably due to YPK inhibition, but Las17 is lost from cortical actin patches (Fig. 5a) and shifts to DAPI-positive mitochondrial clusters (Fig. 5d, e) distinct from the nucleus. Pan1 does not show the same relocalization (Fig. 5b, Supplementary Fig. 2a). We integrated and quantified potential Las17-GFP signal in the nucleus by using DAPI to define the nuclear sphere, yet there was little or no increase in nuclear Las17-GFP signal (Fig. 5f) in response to low or even very high dose Zeocin (500 µg/ml).

We thought it possible that fluorescence microscopy was not sensitive enough to detect a small pool of nuclear Las17 under YCS conditions. We, therefore, used a more sensitive assay for the detection of proteins associated with DNA, namely DamID[71]. We fused the catalytic domain of the bacterial Adenine methyl transferase (Dam) to Las17 and expressed it at a low constitutive level, which allows detection of even transient interactions of the Dam fusion protein with the genome through G^mATC detection[71] (see sketch, Fig. 5g, modified from ref. [72]). Adenine methylation at GATC motifs is detected by restricting total genomic DNA with *Dpn1*, which recognizes G^mATC, rendering *Dpn1*-cleavage a highly sensitive surrogate marker of protein-DNA interaction. As controls, we expressed the Dam methylase alone, and Dam fused to a known DNA binding protein, origin binding factor Orc2 (Fig. 5g). Cleavage by methylation-insensitive *Sau3a* is a positive control for GATC site frequency.

Upon the introduction of Las17-Dam in cells, we found no increase in *Dpn1* cleavage either with or without Zeocin (Fig. 5h), and on the contrary, expression of the Las17-Dam fusion produced less *Dpn1* cleavage than Dam alone, suggesting that Las17 sequesters the methylase away from the nucleus. Orc2-Dam and Dam alone both showed significant DNA association, and neither was influenced by Zeocin (Fig. 5h). Probing genomic DNA by Southern blot for the highly repetitive rDNA array confirmed the observations on total genomic DNA (Fig. 5h). Similar results were observed in the presence of TORC2 inhibition. These results reinforce the Las17-GFP localization results and argue against a direct role for Las17 in the nucleus, at least with respect to Zeocin-induced lesions. This suggests that Las17 acts indirectly to protect cells from misregulated BER and DSB induction[35].

## Does Las17 act through mitochondrial respiration or the LINC complex?

The striking colocalization of Las17 with mitochondria, as well as its presence in a small focus near the Spindle pole body (yeast centrosome) under YCS conditions (arrows, Supplementary Fig. 2b), led us to test mitochondrial function and the LINC complex in YCS. The Sun-domain protein Mps3 is a key component of LINC, which links the actin cytoskeleton to the genome through the inner nuclear membrane[31,32] (Supplementary Fig. 2c, d). However, neither mutation of LINC complex components Mps2 or Mps3 (Supplementary Fig. 2d), nor ablation of ATP synthesis in the mitochondria by mutating the F1F0 ATPase (*atp3Δ*; Supplementary Fig. 2e), altered the yeast cell sensitivity to YCS. We conclude, therefore, that active mitochondrial ATP synthesis is not required for YCS, nor is any cytoplasmic-nuclear communication mediated by LINC. Nonetheless, this does not rule out subtle roles for mitochondrial reticulation[73,74], which occurs on LatB and upon Las17 ablation. We note that Apn1, which cleaves the DNA backbone at abasic sites, is found both in mitochondria and nuclei[75].

## Las17 degradation alters the F-/G- actin ratio in favor of G-actin

Previous work argued that the multiple binding sites for G-actin on Las17 might serve to prevent an imbalanced F-/G- actin ratio[39,66], in parallel to its ability to nucleate actin filaments and stimulate Arp2/3-mediated branching. To see if this occurs under YCS conditions, we monitored the number of cells with visible actin filaments under relevant conditions, by staining with fluorescent phalloidin after fixation (see examples of images in Supplementary Fig. 3; quantified in Fig. 6a, b). Yeast was incubated in YCS conditions (CMB or degradation of Las17-AID by IAA, with or without Zeocin), and we scored the presence of actin filaments and bundles. Given that actin filaments are most visible during S phase in budding yeast, we also scored the fraction of budded cells (S phase). All scoring was performed double-blinded by three independent persons, and the combined ratios were plotted (Fig. 6a). Budding index varies slightly under the conditions scored (Fig. 6b), but this is factored out as we calculated a ratio of budded cells with actin filaments over all budded cells. As expected from imaging data, both the degradation of Las17-AID and incubation with CMB strongly reduced the fraction of cells bearing actin filaments,

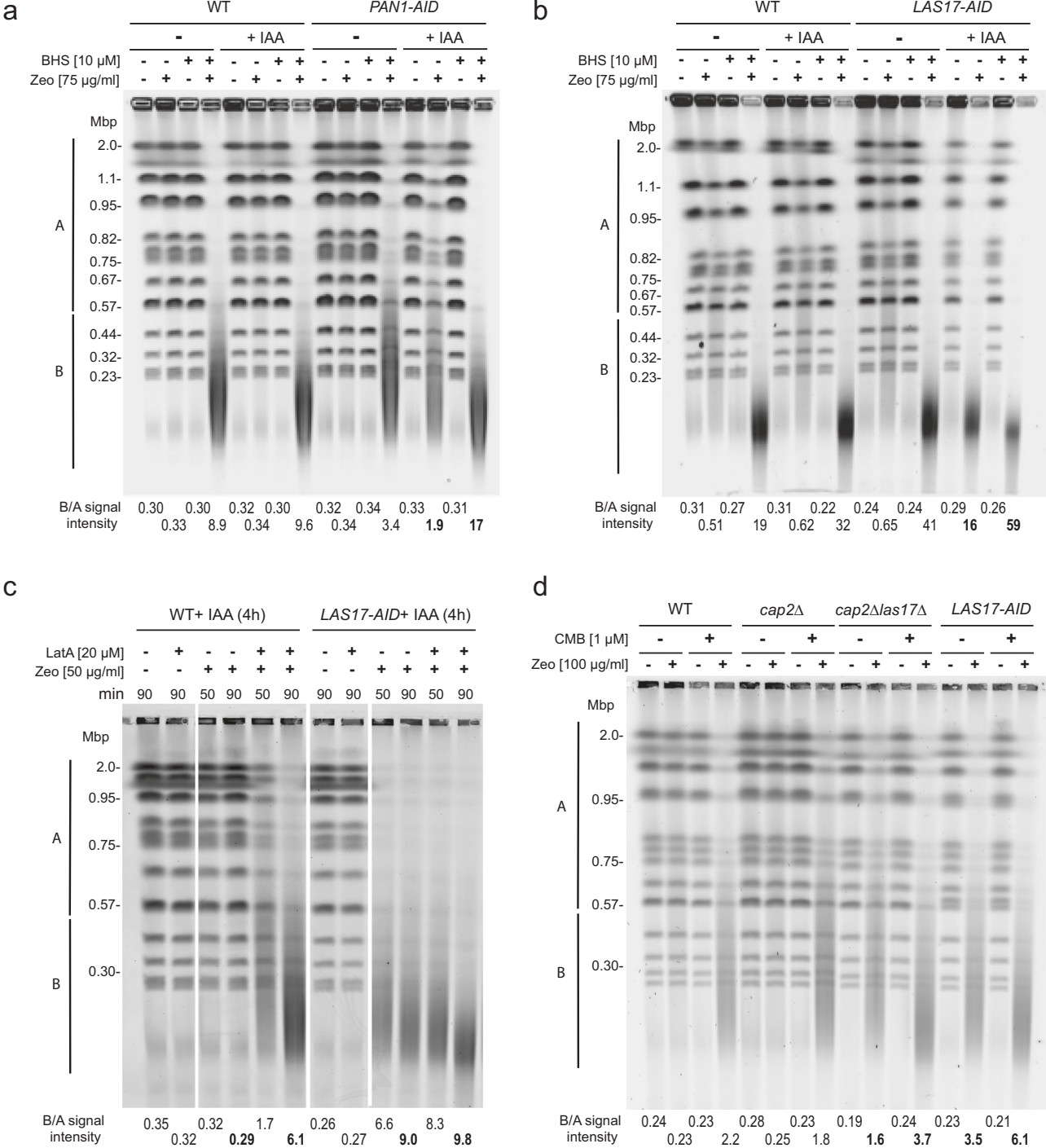

**Fig. 4 | Degradation of Pan1 and Las17 elicit YCS on Zeocin. a** Exponentially growing isogenic WT (GA-5731) and *PAN1-AID* (GA-6810) cells were treated ± 0.5 mM IAA for 4 h to deplete Pan1. Cells were then treated with Zeocin (75 µg/ml) alone or in combination with BHS (10 µM) for 70 min. YCS was monitored as in Fig. 1b and quantitation is described in Methods, with B/A values determined as in Fig. 1 (bold values discussed in text). **b** As **a** but for isogenic WT (GA-5731) and *LAS17-AID* (GA-6839) cells. 0.5 mM IAA was added for 3.5 h and cells were then treated with Zeocin (75 µg/ml) ± BHS (10 µM) for 70 min. YCS was monitored, quantified and scanned as in a. Depletion of Las17 elicits efficient YCS in combination with Zeocin but not alone. **c** Isogenic WT (GA-5731) and *LAS17-AID* (GA-6839) strains were exponentially cultured and treated for 4 h with 0.5 mM IAA for 3.5 h to

trigger Las17-AID degradation as in **b**. Cells were treated with LatA (20 µM), Zeocin (50 µg/ml), or the combination in the presence of IAA for 50 or 90 min as indicated. YCS was monitored and quantified as in **a**. **d** Isogenic WT, GA-4732 (BY4741 WT), GA-10701 (*cap2Δ*), GA-10905 *cap2Δ las17Δ*), and GA-9204 (*LAS17-AID, TIR1*) were cultured in SC overnight, cells were diluted to $OD_{600}$ 0.16 in 20 ml SC and cultured for 3 h with 0.5 mM IAA for 2 h to deplete Las17. Then equal aliquots were treated with 0.25% DMSO (control), 100 µg/ml Zeocin, 1 µM CMB, or the combination of both for 90 min. YCS was monitored, quantified and scanned as in **a**, with SYBR safe staining. The strain background is S288C, not W303, and therefore slightly higher concentrations of reagents were used. The *cap2Δ* mutation partially suppresses YCS provoked by Las17 degradation or by TORC2 inhibition.

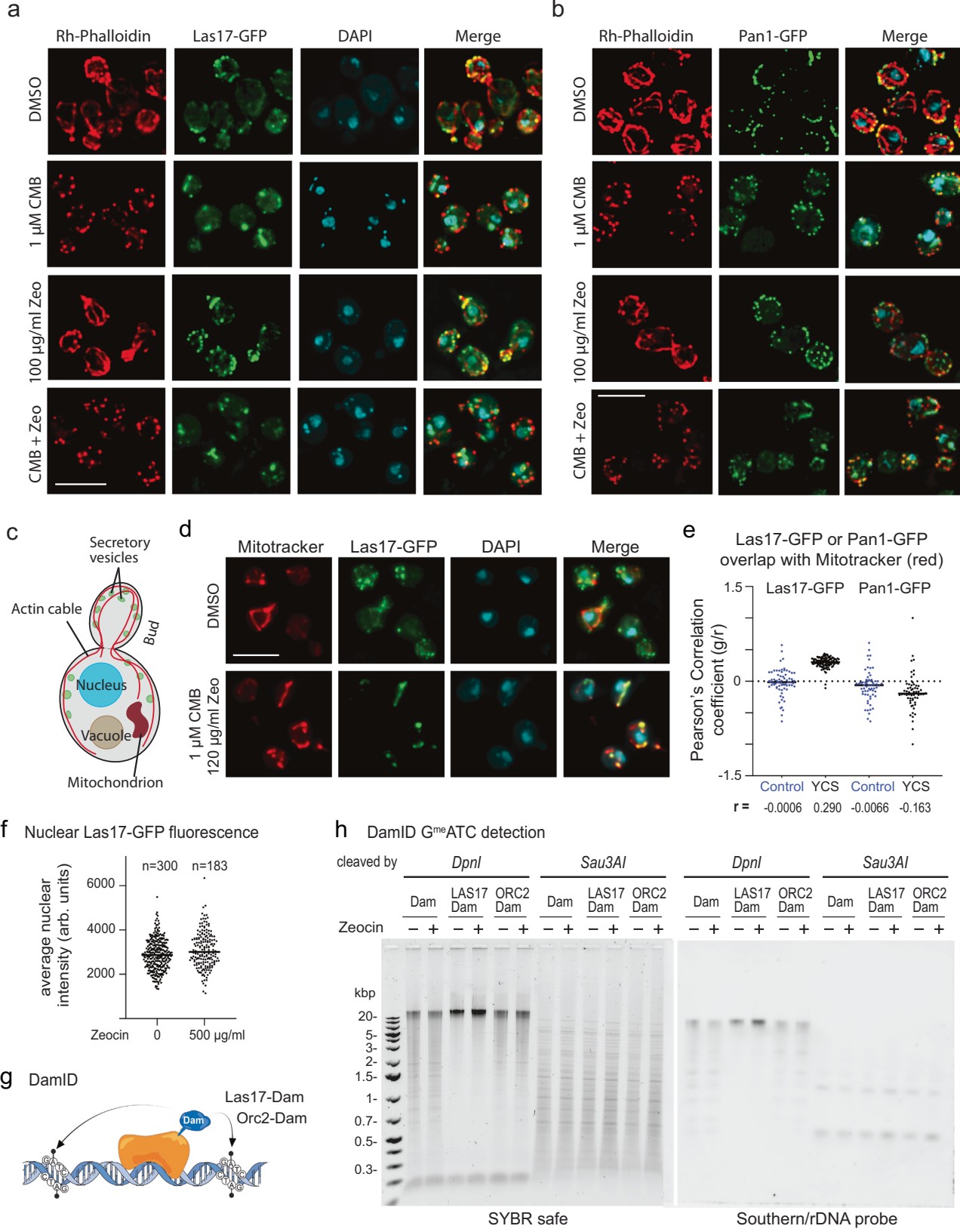

with or without Zeocin (Fig. 6a). To deduce whether actin was degraded or converted to G-actin, we measured overall actin levels in two ways. First, because Cofilin binds both F- and G-actin, we used a previously described, fully complementing Cof1-RFP allele[76] to monitor actin levels under YCS-favoring conditions (Fig. 6c; sample images in Supplementary Fig. 3a, c). We also probed the total actin levels in a

fixed number of cells by western blot, using tubulin as a loading control (Fig. 6d).

In contrast to actin filaments, total actin levels do not decrease upon Las17 degradation based on Cof1-RFP signal, nor in wild-type cells incubated in CMB, either with or without Zeocin (Fig. 6c, d). Given that actin filaments were reduced 3- to 5-fold (Fig. 6a), but overall actin

**Fig. 5 | Pan1 and Las17 do not shift to the nucleus in the presence of Zeocin or CMB + Zeocin. a** Yeast expressing Las17-GFP (GA-6804) was treated 1.5 h with CMB (1 μM), with Zeocin (100 μg/ml) or both. After fixation, DAPI (blue) and F-actin (Rh-phalloidin, red) were captured by spinning disk confocal microscopy. Images are maximum-intensity projections of focal stacks acquired in each channel (see Methods). Bar = 5 μm. **b** Yeast expressing Pan1-GFP (GA-6764) was treated and stained as in **a**. DAPI alone is not shown. Bar = 5 μm. **c** Scheme of a budding yeast cell illustrating structures of interest. **d** Las17-GFP expressing cells were treated as in **a**, but stained with 20 nM Mitotracker Red CMXRos (Thermo Scientific) and DAPI (blue). Colocalization of Las17-GFP and mitochondria is quantified in **e**. Bar = 5 μm. **e** Yeast expressing Las17-GFP or Pan1-GFP (see **a**, **b**) were split and treated 1.5 h either with DMSO (Control, blue) or 1 μM CMB and 150 μg/ml Zeocin (YCS, black). Colocalization of Las17-GFP or Pan1-GFP (green) and Mitotracker (red) was quantified by determining the Pearson correlation coefficient in each single plane of an image stack. n = 118 nuclei for Las17-YCS, all others n = 78; line = median. r values

show a robust correlation of Las17 with mitochondria. Repeated twice. Quantitations in Source Data file. **f** Las17-GFP is not enriched in nuclei on Zeocin. Las17-GFP (green) expressing cells were treated ± 500 μg/ml Zeocin for 1 h at 30 °C, fixed and counterstained with DAPI (blue). Image stacks (green and UV) were analyzed for nuclei spanning at least 5 planes. Nuclear GFP intensity was measured per plane and averaged across ≥5 planes for mean intensity per nucleus, plotted as arbitrary GFP units. n = 300 nuclei without Zeo, mean and S.D. = 2861±700; n = 183 with Zeo, mean and S.D. = 3002±805. Unpaired two-tailed T test with Welch's correction, p = 0.051. See Source Data Files. **g** Concept of DamID mapping[71] of Las17- and Orc2-Dam fusions, derived from an existing sketch[72]. **h** Wild-type (GA-1981) cells expressing Dam, Las17-Dam, or Orc2-Dam were treated ±300 μg/ml Zeocin for 1 h at 30 °C. Genomic DNA digested with DpnI (cleaves at G$^m$ATC) or Sau3AI (cleaves GATC, methylation indifferent), was run on 1% agarose gels; stained with SYBR safe or rDNA by Southern blot (see Methods).

signal does not change, we conclude that G-actin levels increase as actin filaments disappear. In other words, Las17 degradation and/or exposure to CMB increases the balance of G-actin over F-actin.

To determine whether G-actin remained cytoplasmic or became enriched in the nucleus as Las17-AID was degraded, we monitored single confocal planes of Cof1-RFP signal, and integrated this signal within a nuclear mask, generated from the DAPI staining (Fig. 6e, sample images at left). Quantitation showed a significant increase in nuclear actin upon Las17 degradation (Fig. 6e, p < 0.001). This points to an important role of Las17 in regulating the G- and F-actin balance, consistent with evidence that its polyproline-arginine rich domains bind multiple G-actin molecules weakly, while its WH2 domain provides high-affinity G-actin binding[39,66]. If we truncate Las17 at aa 295, just upstream of this polyproline domain, the residual protein should still interact with Arp2/Arp3 but fail to buffer actin[39]. Consistent with the notion that G-actin binding by Las17 protects wild-type cells from YCS on Zeocin, the las17(1−295) allele rendered cells highly sensitive to low levels of Zeocin (Supplementary Fig. 1f).

## The act1-111 allele leads to CMB-insensitive Zeocin-induced breaks

We reasoned that if actin is the culprit that triggers YCS on low-dose Zeocin, then artificially increasing nuclear actin might also provoke YCS. This is indeed the case, as shown in Shimada et al.[35], the ectopic expression of actin mutants lacking the nuclear export signal (act1$^{nes}$) provoked weak but detectable chromosome fragmentation, even without TORC2 inhibition (B/A ratios 1.4 to 2.3; see ref. 35). This was shown not only for wild-type actin$^{nes}$, but for actin forms that either favor or impair filament formation (act1-S14C and act1-A204E/P243K, respectively), as well as one filament-forming mutant that is impaired in lateral protein-protein interactions (act1-111[77,78]). This non-lethal actin mutation, act1-111, has an altered acidic patch on the lateral side of the molecule (Fig. 7a; point mutations D222A, E224A, and E226A[78]), which slows down endocytosis and cell division due to the partial dissociation of actin cables[79] and an inability to bind the Las17 polyproline domain[39]. Nonetheless, act1-111 protein remains at wild-type levels (Fig. 7b, c) and the act1-111 allele was hypersensitive to Zeocin (at 20 μg/ml, Fig. 7d), reminiscent of the Zeocin sensitivity observed for the Las17 truncation that eliminates its G-actin-binding domain (Supplementary Fig. 1f). Importantly, act1-111 sensitivity to Zeocin was fully complemented by introduction of a single copy wild-type pACT1 plasmid (Fig. 7d).

To see if act1-111 affects YCS, the mutant strain expressing act1-111, either complemented or not by +pACT1, was screened for chromosome fragmentation on Zeocin, in the absence and presence of CMB (Fig. 7e, f). The act1-111 protein, rather than enhancing YCS as it did when it accumulated in the nucleus due to mutated export signals[35], suppressed fragmentation in the presence of either CMB or LatA (Fig. 7e, f; bold values; B/A = 0.87 vs 8.1 for wild-type on CMB). This

resistance was overcome by expressing wild-type ACT1 (+pACT1, Fig. 7e, f). In contrast, another non-lethal actin mutant, act1-129 (R177A, D179A; green in Fig. 7a)[77], which allows actin assembly into cables, patches, and microfilaments, but fails to bind certain proteins such as Abp1, was only weakly compromised for YCS (Fig. 7f; B/A = 7.2 or 6.5 vs 2.5 for act1-111).

The act1-111 mutation does not alter Las17 distribution under YCS conditions (i.e. CMB + Zeocin, Supplementary Fig. 4), allowing Las17 association with mitochondria and no visible enrichment of Las17 in nuclei, resembling wild-type cells. Since Las17-GFP co-localizes with mitochondria even under conditions that suppress chromosome fragmentation (cf. act1-111 vs. act1-111 + pACT1; Supplementary Fig. 4), we can conclude that the shift of Las17 to mitochondria is not sufficient to trigger YCS.

## The actin-containing chromatin remodeler INO80C is required for YCS

Besides interfering in Las17 binding, the residues mutated in act1-111 (D222A, E224A, and E226A) are found at the binding interface of actin and Arp8, an integral component of the multisubunit INO80C remodeler (Supplementary Fig. 5[80,81]). The mutations destabilize an important alpha helix that forms salt bridges to Arp8. Whereas G-actin forms a heterodimeric complex with Arp4 in chromatin multiple modifying complexes (notably, INO80C, SWR-C, and NuA4[12,14,44]; Fig. 8a), its interaction with the INO80C-specific subunit Arp8, creates an Arp8-Arp4-actin-Ies4 module that binds the HSA domain of the Ino80 catalytic subunit (aa 356−691[81]). This subcomplex plays a central role in the ability of INO80C to bind its substrate nucleosome during remodeling reactions[13,80,81]. Knowing that INO80C remodeling activity promotes the processivity of DNA replication polymerases, both in recovery from fork stalling[40,41,43] and in an unchallenged S phase[45], we next asked whether the changes in G-/F- actin ratio provoked by Las17 degradation and TORC2 inhibition, influences long-patch BER through nucleosome remodeler activity[35]. We note that the study of nucleosome remodeler mutants, and especially ino80 mutants in yeast, requires the study of more than one background, as ino80Δ is lethal in W303-derived strains, but viable in the S288C background[44,82]. Moreover, the two backgrounds have different sensitivities to some DNA-damaging agents.

We performed the YCS assay on mutants of core subunits of the INO80, ISW1/ISW2, SWR-C, NuA4, and CHD1 remodelers, under two different concentrations of Zeocin and TORC2 inhibitor and, where necessary, in both the W303 and S288C (BY) backgrounds. We tested the actin-independent Isw1 and Isw2 remodelers because they have been proposed to act on DNA replication polymerase processivity in a chromatin context[42,43,83]. We found that the isw1Δ isw2Δ double and the chd1Δ single mutant was actually more susceptible to YCS than the isogenic wild-type W303 strain, whereas the arp8Δ mutant, which significantly reduces INO80C activity and fork processivity[40,41,43] was

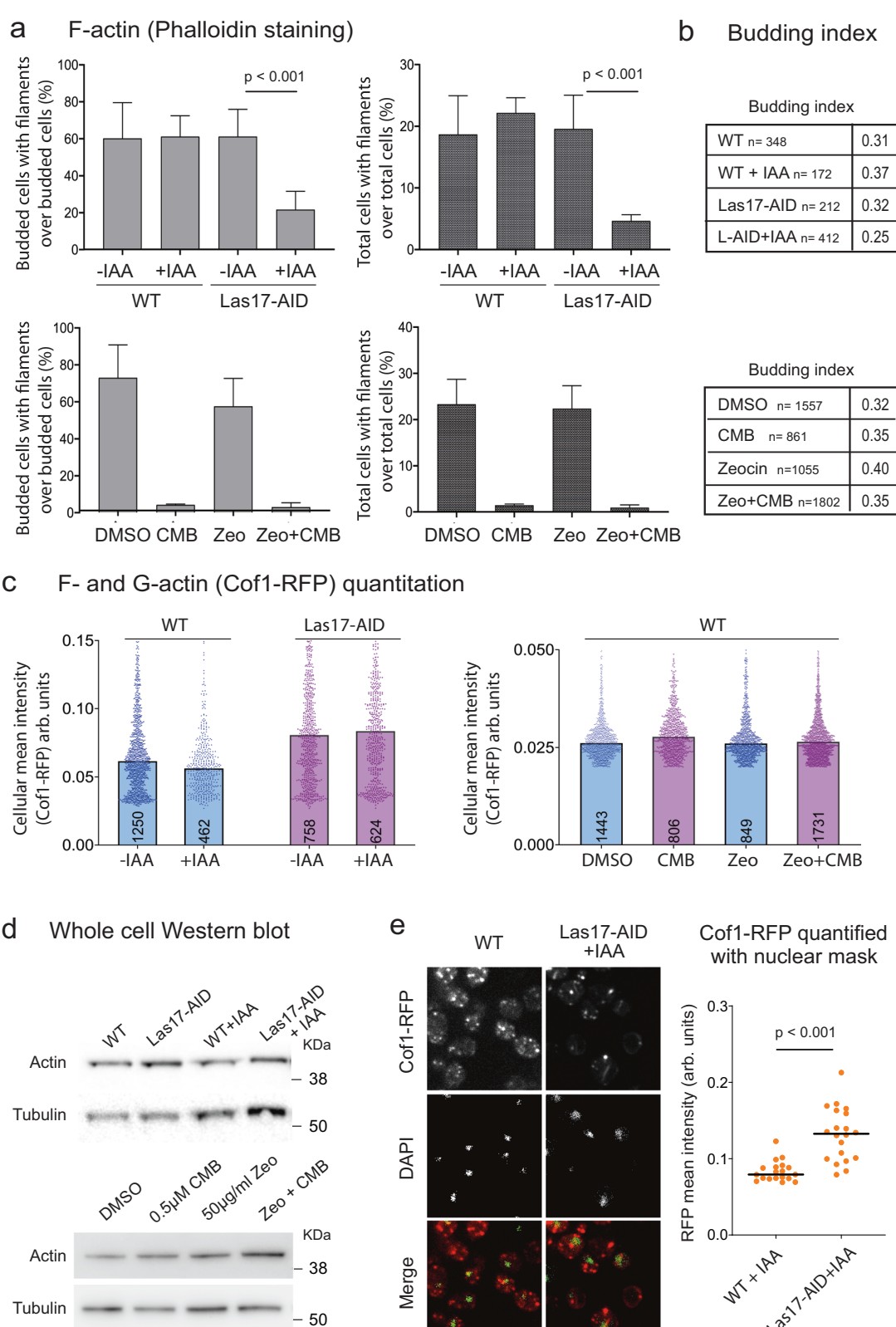

a  F-actin (Phalloidin staining)

b  Budding index

| Budding index | |
| --- | --- |
| WT n= 348 | 0.31 |
| WT + IAA n= 172 | 0.37 |
| Las17-AID n= 212 | 0.32 |
| L-AID+IAA n= 412 | 0.25 |

| Budding index | |
| --- | --- |
| DMSO n= 1557 | 0.32 |
| CMB n= 861 | 0.35 |
| Zeocin n=1055 | 0.40 |
| Zeo+CMB n=1802 | 0.35 |

c  F- and G-actin (Cof1-RFP) quantitation

d  Whole cell Western blot

e

partially resistant to shattering (B/A = 4.14 vs 21.8, *arp8Δ* vs. wild-type on 50 μg/ml Zeo + CMB; Fig. 8b). This resistance was less pronounced at higher Zeocin levels, but was also observed in the S288C-derived BY background (Fig. 8d). It is not clear why the *isw1Δ isw2Δ* double mutant is more susceptible to YCS, but the fact that the triple *isw1Δ isw2Δ chd1Δ* showed weak resistance (B/A = 6.6 vs 21.8, on 50 μg/ml Zeo + CMB, Fig. 8b) may reflect altered nucleosome spacing and changes in

lesion accessibility in the mutant[84]. The resistance we observe in *arp8Δ* cells is consistent with data showing that INO80C promotes DNA polymerase δ and ε processivity[40,41,43], as both polymerases have been implicated in long-patch BER[35]. Our finding that resistance drops at higher doses of Zeocin is also consistent with this model, as the higher the density of oxidative lesions, the less impact polymerase processivity should have[35].

**Fig. 6 | Las17 degradation and TORC2 inhibition decrease F-actin altering the G-/F-actin ratio. a** Fraction of budded cells or total cells with F-actin filaments was determined after staining with Rh-phalloidin. Exponentially growing cells (GA-5731, AID control; GA-6840, Las17-AID; GA-1981, WT) were treated as indicated (±0.25 mM IAA for Las17-AID and its control, and 50 µg/ml Zeocin ± 0.5 µM CMB for WT), then fixed and stained with Rh-phalloidin. Image examples in Supplementary Fig. 3. Both brightfield and fluorescent images were scored for budding index and cells with filaments by three independent operators on blinded images. Number of cells scored per condition is shown in **b**. Plots show % budded cells with filaments (left) or % total cells with filaments (right). Data presented as mean value (top of bar) ±SEM. Raw numbers in Source Data file. **b** Budding index (bud presence) was scored for the same cell populations analyzed in **a** by three operators on blinded images. Total number cells are same for **a**, **b**. Plotting Budded cells with filament/ Budded cells excludes cell cycle impact on results. **c** Yeast strains as **a** were

transformed with a Cof1-RFP (internal tag, Addgene #37102) expression plasmid and treated as above. Cof1-RFP intensity monitors both G- and F-actin and was quantified for each condition with a CellProfiler pipeline. Each dot represents one cell and the number of cells scored range from 402 to 1731, as indicated within each bar. All data points are plotted and top of bar indicates median. By Wilcoxon rank sum test actin levels do not vary significantly ± IAA nor under YCS conditions. **d** Cells of the samples used in **a** were collected before fixation. Total protein extracts were subjected to SDS-PAGE and western blotting with validated anti-actin (My BioSource, BSS9231831) or anti-tubulin (Abcam, ab6161). **e** Scoring nuclear actin detected through Cof1-RFP (expressed as in **c**) using nuclear masking based on DAPI staining (Fig. 5f) on WT and an isogenic strain lacking Las17 (LAS17-AID + IAA). Samples visualized as **c**. Bar = 10 µm. In graph of mean (bar = mean values) of $n$ nuclei ($n$ = 19, WT; Mean = 0.8339 ± 0.0131; $n$ = 20 Las17-AID; Mean = 0.1316 ± 0.0340). Based on a two-tailed $T$ test $p < 0.0001$. Data in Data Source file.

With respect to other actin-containing remodelers, deletions of SWR-C subunits (Bdf1) or NuA4 subunits (Eaf1 or Eaf7) showed no pronounced resistance to YCS over the wild-type BY background at the lower Zeocin concentration (80 µg/ml Zeo + CMB, Fig. 8c). Importantly, despite a partial resistance to YCS observed for the *arp6Δ* allele (B/A = 3.3 vs 7.1, Fig. 8c), ablation of the catalytic subunit of the SWR-C complex, *swr1Δ*, led to enhanced YCS while loss of its ATPase activity had no effect (Supplementary Fig. 6a). Similarly, the deletion of *SNF2* and *SWI3*, which encode subunits of the Snf2/Swi remodeler, failed to alter YCS efficiency (Supplementary Fig. 6b).

In the *arp8Δ* background the INO80 complex lacks a key structural module (Arp8, Arp4, actin, Ies4), and has a decreased (but not abolished) rate of remodeling[7]. We, therefore, asked whether YCS in *arp8Δ* yeast might be delayed, rather than abolished, by testing YCS after 45 min and 90 min exposure to CMB and Zeocin (Fig. 8d). Indeed, resistance to YCS conditions was more pronounced at 45 min (relative to the *ARP8+* strain) than at 90 min (B/A = 0.96 vs 3.7 at 45 min; 2.2 vs 6.5 at 90 min; Fig. 8d). Finally, we tested the ATPase-dead INO80 allele, by expressing the DEAQ-box dead *ino80-K737A* mutant in the *ino80Δ* BY background. We found that the K737A ATPase mutant was resistant to YCS at high dose Zeocin (B/A = 3.1 vs 13.0 for the WT background; Fig. 8e). Together, these results suggest that the INO80C remodeler facilitates YCS under conditions that increase G-actin levels in the nucleus. Importantly, without CMB the *ino80* and *arp8* mutants are indistinguishable from wild-type in their response to Zeocin with respect to chromosome fragmentation (B/A = 0.78, 0.74, 0.63, 0.67; Fig. 8d).

## Discussion

The regulation of polar cell growth, sphingolipid biogenesis, and the actin cytoskeleton[48] by TORC2 is conserved from yeast to man. With respect to the dramatic fragmentation of the yeast genome that is triggered by TORC2 inhibition in the presence of Zeocin (YCS)[34,35], an analysis of mutants argues that the fragmentation is due to altered long-patch BER, through which clustered base oxidation events are misprocessed to generate DSBs[35]. We have narrowed the window of interference to either Apn1 access or DNA polymerase processivity, which are both relevant for repairing clustered lesions such as those induced by bleomycin-like compounds, including Zeocin[46]. Appropriate repair requires tight coordination of enzymatic processing so that one lesion is re-ligated prior to Apn1-mediated cleavage of an adjacent one. An enhanced rate of DNA polymerase translocation can drive the repair on one strand into a nick on the opposite strand, generating a DSB (Fig. 9; and ref. 35). Coordinated BER at clustered sites of damage has been reported to occur in species ranging from bacteria[85,86] to man[87,88], but the mechanisms of control have remained obscure.

Here, we have systematically analyzed the upstream trigger, that is, the pathway that links TORC2 inhibition to misregulated LP-BER.

First, we confirm that the maintenance of cytoplasmic F-actin, and controlled levels of nuclear G-actin, are necessary to ensure appropriately timed repair of clustered base lesions in yeast. Actin depolymerization triggered by latrunculin A (or LatB and cytochalasin D in human cells) has a similar effect as TORC2 inhibition with respect to Zeocin-induced YCS (Figs. 1 and 2). By testing actin regulatory factors that show altered phosphorylation patterns during YCS, we found that the depletion of the actin chaperone Las17, homolog of human WASP, most closely phenocopied TORC2 inhibition on Zeocin (Fig. 4; Supplementary Fig. 1). The induced degradation of Las17, like the inhibition of TORC2, alters the G-actin: F-actin balance, notably leading to an increase of G-actin the nucleus (Fig. 6). In addition, YCS can be induced weakly by expressing a nuclear export deficient actin[35]. This fact, combined with the epistasis observed for LatA and Las17 degradation (Fig. 4c), and the suppressive effects of *cap2Δ* (Fig. 4d), argues that depolymerized cytoplasmic actin filaments−if not buffered by Las17 binding of G-actin−leads to elevated levels of nuclear actin that promote DSB formation during the repair of clustered lesions by BER (Fig. 9).

Contrary to expectations based on mammalian systems, we are unable to detect Las17, the yeast WASP homolog, in the nucleus after Zeocin treatment (Fig. 5). Nor do we detect actin filaments in the yeast nucleus. The fact that a truncated form of Las17 that removes its G-actin-binding domains renders cells hypersensitive to Zeocin (Supplementary Fig. 1f), leads us to propose that Las17 normally buffers a G-actin pool through its polyproline domain as well as other actin-binding domains[39,65], to prevent a pathological accumulation of nuclear actin during fluctuations in the cytoplasmic actin cytoskeleton.

### Nuclear actin alters repair dynamics through the remodeler INO80C

The question remained how nuclear actin levels influenced the conversion of Zeocin-induced based damage into lethal DSBs (Fig. 9). Here, we show that the *act1-111* mutant, which carries mutations interfering with a lateral protein-binding interface, forms actin filaments and supports cell growth[77], yet is highly toxic on Zeocin (Fig. 7d). The *act1-111* mutation alone triggers a low level of chromosome fragmentation on Zeocin, presumably due to impaired BER, but at the same time, confers resistance to the high-level YCS triggered by TORC2 inhibition or LatA (Fig. 7e, f). Elsewhere it was reported that the act1-111 protein fails to bind the arginine/polyproline domain in Las17[39], yet it is also predicted to compromise Arp8-actin interaction within the INO80C complex[13,80,81] as shown in Supplemental Fig. 5. The results presented here argue that act1-111 and INO80C influence the efficiency of YCS, most likely by altering BER regulation.

A number of previous studies in mammals and flies have proposed active engagement of WASP and nuclear actin in DNA repair. We note that none of these has tested whether actin-dependent nucleosome remodelers are relevant for the observed phenotypes (reviewed in

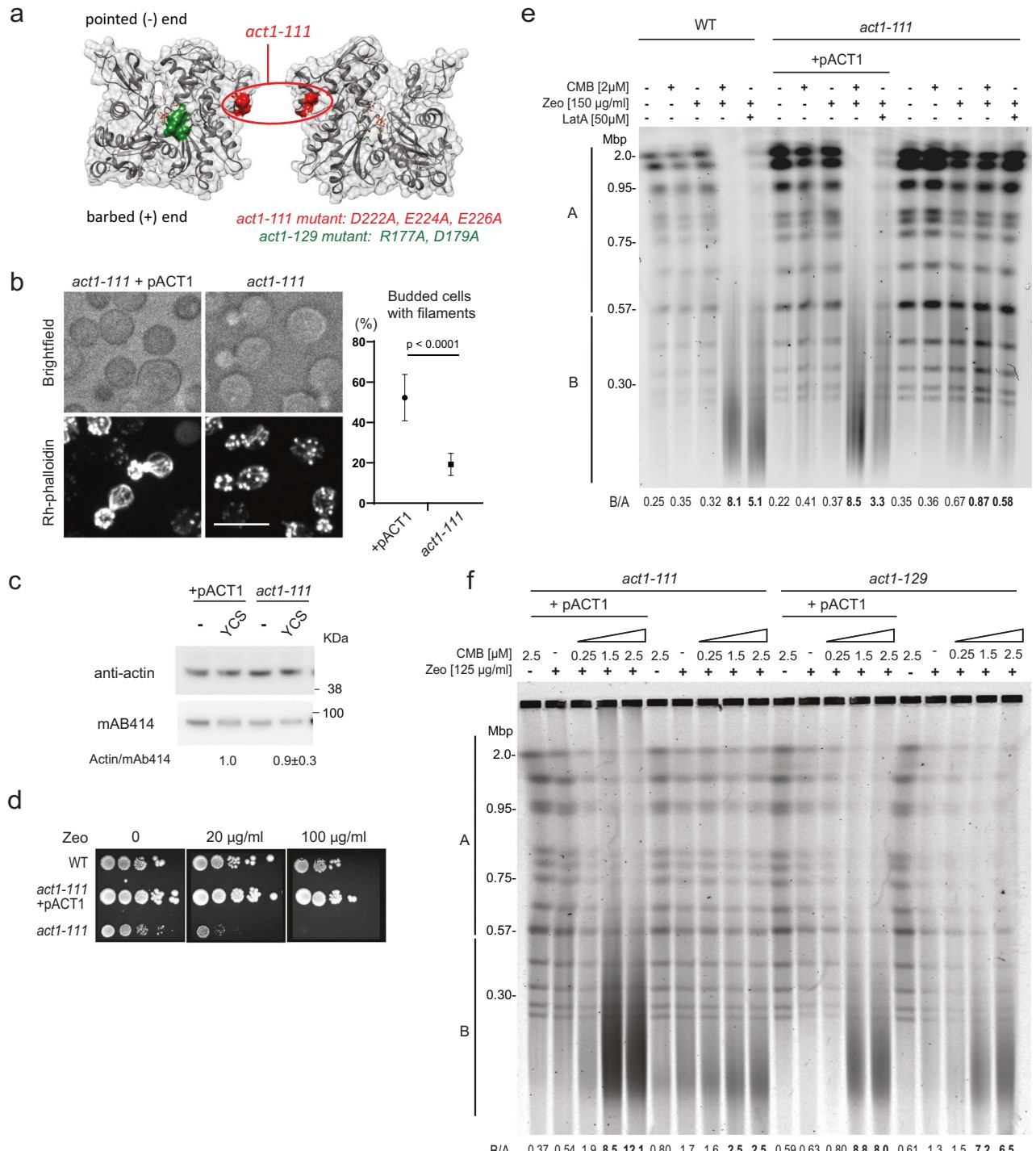

**Fig. 7 | The *act1-111* mutation attenuates TORC2-Zeocin-induced YCS. a** Position of *act1-111* (red) and *act1-129* (green) mutations[77] on an actin monomer shown in two orientations. Filament formation is intact through pointed and barbed ends. **b** Actin cytoskeleton staining in *act1-111* (GA-8592) ±pACT1. Exponentially growing cells were fixed with PFA, stained Rh-phalloidin for 2 h, and imaged. The phalloidin signal in *act1-111* expressing pACT1 resembles wild-type, while *act1-111* has mostly cortical F-actin patches. Quantitation of actin cables was scored by two blinded operators on three isolates of the *act1-111* mutant vs *act1-111* + pACT1. *n* total +pACT1 = 1012, *n* total *act1-111* = 2790; data are presented as mean value ±SEM, by two-tailed unpaired *T* test, *p* < 0.0001. Scoring is in Source Data file. Scale bar = 10 μm. **c** Actin levels are constant under YCS conditions in *act1-111* strains. Whole protein extracts from strains grown as in **b** were analyzed by western for a nuclear pore protein (Mab414, Abcam) and actin (My BioSource). Quantitation reflects four blots scanned by the Typhoon scanner; Actin:Mab414 value on YCS is normalized to

+pACT1. **d** The *act1-111* mutant (**a**) is hypersensitive to Zeocin. 1:10 dilution series of *act1-111* ± pACT1 on SC agar plus glucose with indicated Zeocin concentrations. Colonies grew 3 days at 30 °C; performed in triplicate with similar results. **e** Las17-GFP *act1-111* cells (GA-8592) ± pACT1 plasmid are resistant to YCS. An isogenic wild-type background (GA-9247; BY background) and the *act1-111* mutant were treated with 2 μM CMB and 150 μg/ml Zeocin for 1.5 h, and genomic DNA was monitored by CHEF gel analysis. Where indicated, 50 μM LatA was added rather than CMB. Gels and quantitation are as Fig. 1b. B/A values show that *act1-111* is resistant to CMB and LatA treatment. Experimental repeats (*n* = 3) include panel f. **f** Wild-type and *act1-111* (GA-8592) and *act1-129* cells (GA-8590) ± pACT1 plasmid were treated 1.5 h with Zeocin (125 μg/ml), a titration of CMB (0.25–2.5 μM) or Zeocin and CMB as indicated. Genomic DNA was monitored by CHEF gel, stained, and quantified as Fig. 1b. The *act1-111*, but not *act1-129* cells show less CMB-induced fragmentation. The experiment was repeated three times.

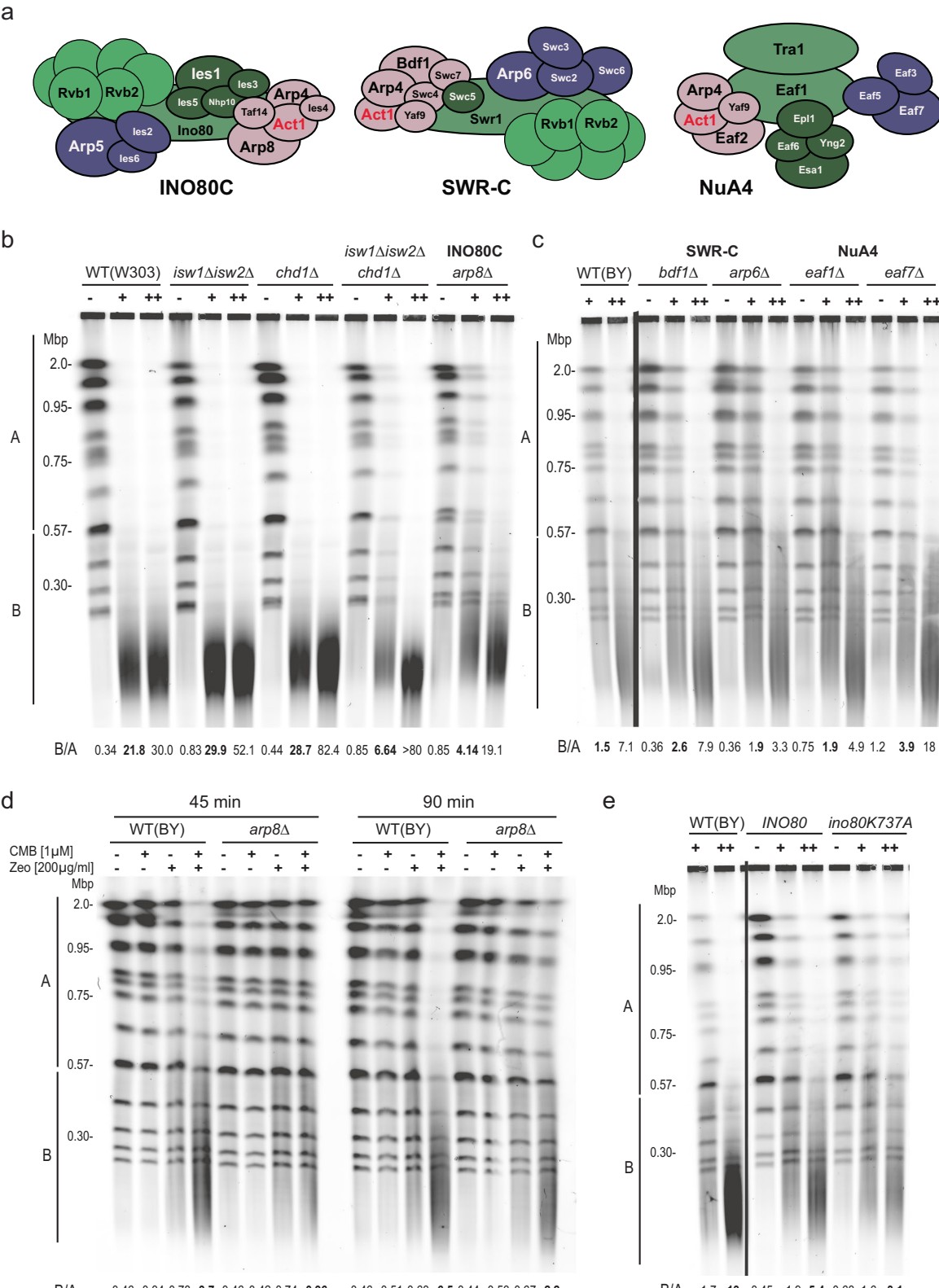

ref. 17). We have shown that a reduction in INO80C activity leads to partial resistance to the conversion of oxidative damage to DSBs, triggered by TORC2 inhibition. This argues that the INO80C remodeler plays a positive role in the conversion of adjacent base lesions to DSBs, possibly by its well-established role in promoting DNA polymerase processivity[40,41,43,45] (notably of DNA Pol δ and Pol ε)[35], which are important in LP-BER. Changes in nucleosome spacing may also

enhance YCS, by increasing access for Apn1 to cleave adjacent lesions in an untimely manner. Despite the redundancy that occurs among remodeler complexes (e.g., INO80C, ISW1, ISW1, and CHD1) for some processes, our genetic data implicates the action of INO80C in the misregulation of LP-BER that leads to DSB formation (Fig. 9).

Intriguingly, two actin-related subunits of INO80C, Arp8, and Arp4, cooperate to inhibit actin polymerization[89], presumably by sequestering

**Fig. 8 | Functional INO80C is required for efficient YCS. a** Sketches of subunit composition of the *S. cerevisiae* INO80C, SWR-C and NuA4 chromatin remodelers and modifiers each containing actin as a core subunit[12–15]. Subunit names are from budding yeast. **b** Individual ISWI and CHD remodelers are not needed for YCS, while loss of Arp8, a unique INO80C subunit confers partial resistance. Exponential cultures of wild-type (WT) W303 (GA-9875); *isw1Δ,isw2Δ* (GA-9878); *chd1Δ* (GA-9879); *isw1Δ isw2Δ chd1Δ* (GA-9882) and *arp8Δ* (GA-9848) were treated for 1.5 h with 0.5 μM CMB and 50 μg/ml Zeocin (+) or 1 μM CMB and 100 μg/ml Zeocin (++), prior to processing for CHEF gel analysis. Quantitation as in Fig. 1b. **c** Isogenic strains in the BY background lacking indicated remodeler subunits of SWR-C or NuA4 were tested in the YCS assay. Note that this background is more resistant to Zeocin than W303. BY4733 (GA-2263), *bdf1Δ* (GA-9953), *arp6Δ* (GA-2319), *eaf1Δ* (GA-9954), *eaf7Δ* (GA-9955) were grown to exponential phase and treated for 1.5 h with DMSO (−), 0.5 μM CMB and 80 μg/ml Zeocin (+) or 1 μM CMB and 125 μg/ml Zeocin (++),

prior to CHEF gel analysis. Quantitation as in Fig. 1b. The *eaf7Δ* mutant showed slight resistance at low concentrations of Zeocin/CMB, while *arp6Δ* and *eaf1Δ* mutants showed minor resistance at the higher concentration. Except for INO80C, the minor effects were inconsistent for a given remodeler. **d** Deletion of *arp8* delays YCS consistent with a requirement for functional INO80C for polymerase processivity during LP-BER. The indicated isogenic strains (*ARP8* + = GA-9247; *arp8Δ* = GA-9250) were incubated with the indicated compounds for either 45 or 90 min, prior to processing for CHEF gel analysis. Quantitation is as in Fig. 1b. At 45 min *ARP8*+ started to show fragmentation, unlike *arp8Δ*. The experiment was repeated twice. **e** The ATPase mutant in the catalytic subunit ino80 (K737A) confers resistance to YCS. Performed in the BY background, *ino80Δ* (GA-2264) cells bearing Cen-plasmids expressing WT INO80 or *ino80 K737A* were treated for 1.5 h with 0.5 μM CMB and 80 μg/ml Zeocin (+) or 1 μM CMB and 125 μg/ml Zeocin (++); CHEF gel analysis and quantitation as Fig. 1b. Experiment repeated twice.

actin as a monomer within the remodeler. Given that the Arp8 module of INO80C leaves the pointed end of actin exposed[80], it has been proposed to be able to alter actin polymerization[90,91]. Thus, although we favor the model that INO80C acts through nucleosome displacement to promote polymerase processivity, it might also modulate actin dynamics.

### Linking the TORC2-Ypk1/2 pathway to chromatin
We have systematically explored and eliminated other hypotheses as to how the TORC2-Ypk pathway can alter BER, such as through LINC, mitochondrial function, or by forming nuclear actin filaments. Whereas phosphoproteomics is neither exhaustive nor highly sensitive, it successfully identified a group of proteins that work together to regulate cortical actin filament growth in response to YCS conditions, including Las17. Although we failed to detect altered phosphorylation on Las17-S554, a site thought to repress high-affinity actin binding by the Las17 WH2 domain[66], we did detect an increase in phosphorylation of T380, which lies between two conserved polyP/R domains that bind G-actin with low affinity[39,66]. It may be that multiple phosphorylation events are needed to alter Las17-actin interaction. The most pronounced effect on YCS was, of course, by *LAS17-AID* degradation (Figs. 3 and 4).

Theoretically, an alternative mechanism through which actin dynamics might regulate BER is the release of rate-limiting BER factors from cytoplasmic actin filaments (e.g., Apn1). There is precedent for cytoplasmic F-actin sequestering factors, such as aldolase, an enzyme that is released when the PI3 kinase Rac depolymerizes the actin cytoskeleton[2]. However, YCS is not increased by Apn1 overexpression, nor do nuclear Apn1/Ogg1 levels increase upon F-actin disassembly[35]. Thus, rather than Apn1 release from cytoplasmic actin, we believe its increased or ill-timed cleavage at adjacent lesions is relevant in YCS.

### Conservation of the actin cytoskeleton-BER link in humans
In human cells, the effect of TOR inhibition or actin depolymerization coupled with Zeocin, led to increased damage even when treatments were sequential (Fig. 2b–d). The outcome was not a massive fragmentation of the human genome, yet we could detect an increase in a DNA damage marker (γH2AX; Fig. 2a, c) and in the number of DNA nicks and breaks based on the Comet assay (Fig. 2d). The synergistic impact of actin depolymerization on cell survival is particularly pronounced in MLH1-deficient HCT116 cells, consistent with evidence that actin cytoskeleton disorganization can contribute to oncogenesis[6–8]. The fact that there is no rapid conversion of oxidative damage to DSBs in the human cells, as opposed to yeast, may be due to their robust short-patch BER system that makes use of a nonprocessive polymerase (DNA pol β), and dedicated factors, such as XRCC1, PARP and Ligase III[92]. Nonetheless, given that oxidized bases are processed to abasic sites that are cleaved by APE1 in both short- and long-patch BER in mammals, we expect that the repair of clustered lesions is as tightly regulated in humans as in yeast[87,88]. It may be simple redundancy of pathways (SP- and LP-BER and chromatin modulating) that renders

human cells less sensitive to TORC2 inhibition and Zeocin. Nonetheless, we expect that mammalian BER is also sensitive to actin-dependent chromatin remodelers (e.g., human SWI/SNF, SRCAP, and EP400)[12].

The impact of actin depolymerization on repair may be relevant to the therapeutic use of γ-radiation or bleomycin in cancer therapy, as preliminary evidence suggests that actin depolymerizers, like inhibitors of nucleosome remodelers, can be effective in combination therapy to treat some tumors[93]. Moreover, premature aging seems to be influenced by nuclear F-actin[94]. Thus, linking actin-related pathologies in humans to nucleosome remodelers and the repair of oxidative lesions merits further study.

## Methods
### YCS assay
All yeast strains are described in Supplementary Table 2. To induce YCS, yeast cultures were grown in synthetic complete (SC) media with 2% glucose so all reached reached an $OD_{600}$ ~ 0.5 simultaneously. Cultures were split and treated with the indicated concentrations of drugs for the indicated time (ranging from 45 min to 90 min, as indicated). Cultures were washed in ice-cold wash buffer (10 mM Tris-HCl pH 8, 50 mM EDTA), and yeast genomic DNA was prepared for PFGE or CHEF gel analysis. Cells were resuspended in 50 μl Zymolase buffer (100 mM $NaHPO_4$ pH 7, 50 mM EDTA, 1 mM DTT, 0.8 mg/ml Zymolase) and mixed with 50 μl 3% pulse-field certified agarose (Bio-Rad) in 0.5× TBE buffer to form a plug. Plugs were incubated for 60 min at 37 °C in Zymolase buffer (100 mM $NaHPO_4$ pH 7, 50 mM EDTA, 1 mM DTT, 0.4 mg/ml Zymolase), washed in wash buffer and then incubated at 50 °C with Proteinase K buffer (10 mM Tris-HCl, 50 mM EDTA, 1% Na-N-Lauroyl Sarcosinate and 1 mg/ml Proteinase K) overnight. Plugs were washed in wash buffer and subjected to a gel of 1% pulse-field certified agarose (Bio-Rad) in 0.5× TBE. Yeast chromosomes were separated in a CHEF-DRII (Bio-Rad) apparatus at 14 °C with 6 V/cm and a 60-90 s switch time at an included angle of 120° for 24 h.

### Quantitation of chromosome fragmentation
Gel quantitation was performed as described in the accompanying paper[35]. In brief, SYBR safe dye (Invitrogen) or Ethidium bromide fluorescent staining of dsDNA on non-denaturing CHEF gels is captured on the Typhoon FLA 9500 scanner with LPB (510LP) filter (GE Healthcare) or rarely, with the Chem Doc XRS system (Bio-Rad). Both show linearity over $10^4$-fold dilution. For quantitation, the rectangle that covers most of a given lane was set as ROI, and DNA intensity was plotted. Background from a flanking region of the gel without a sample was subtracted. Signal intensity was integrated using line plot profiling, and the integrated value from 0.57 Mbp to 2.2 Mbp, and from 0.56 Mbp to ~50 kbp was used to determine the degree of chromosome fragmentation. There is variation of signal from experiment to experiment, but intact chromosomes give a B/A ratio under 0.5, and significant fragmentation generates B/A ratios >1.5 up to values over

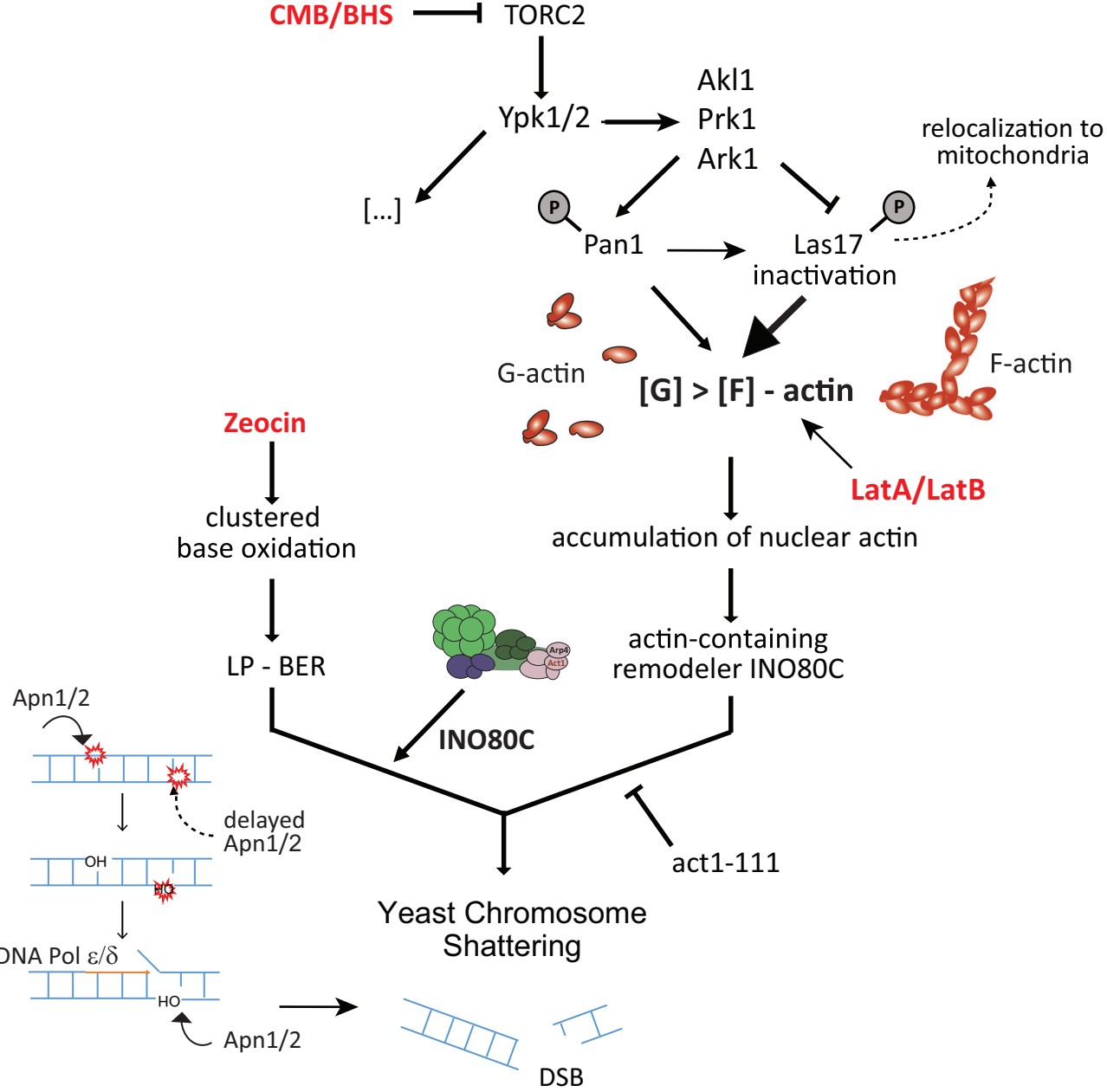

**Fig. 9 | Model for synergistic action of TORC2 inhibition and Zeocin driving fragmentation.** A mechanistic model of how TORC2 inhibition impacts base excision repair to generate rapid and irreversible DSBs in yeast. Related imidazo-quinolines CMB4563 and NVP-BHS345 inhibit TORC2[34,35] and in turn YPK1/YPK2, which inactivates Alk1, Prk1 and Ark1[34,36,37]. These kinases lead to the disruption of several actin regulatory complexes that primarily downregulate endocytosis but also lead to Las17 inactivation and partial relocation to mitochondria. TORC2 inhibition, like the degradation of Las17, leads to increased availability of non-filamentous globular actin (G-actin) and its nuclear accumulation. In Zeocin-treated yeast, the presence of enhanced levels of nuclear actin could activate actin-dependent nucleosome remodelers such as INO80C to increase endonuclease access at paired base oxidation events on opposite strands[35] (as shown) and increase DNA polymerase processivity[40-45]. When the processive replication polymerase meets a single-strand break created by premature Apn1 (or Ogg1) activity, DSBs can form ref. [35]. Our results suggest a role for chromatin in the appropriate coordination of repair of clustered oxidative lesions.

80. For calculation of estimated DSB lesions see ref. [35]. All CHEF gels presented here were quantified for integrity of chromosomal DNA and values are summarized in Supplementary Data 1.

### Trichloric acid (TCA) precipitation, SDS-PAGE, and western blots

2 ml of yeast cells ($OD_{600}$ 0.4–0.8) were pelleted by centrifugation and lysed with 100 µl 80% 2 M NaOH, 20% Thioglycerol for 10 min on ice. Total protein was precipitated by addition of 100 µl TCA (50%) for 10 min on ice and pelleted by centrifugation at 4 °C for 10 min at top speed. Samples were washed with 1 ml 50 mM Tris/HCl pH 8 and boiled for 10 min at 95 °C with 2× western blot sample buffer (NuPAGE).

Proteins in sample buffer containing reducing agent (both from NuPAGE) were size separated in 4-12% gradient Bis/Tris gels (Bolt™) for 40–90 min at 120–200 V with either MOPS/MES SDS running buffer (1x, NuPAGE). Proteins were transferred to a PVDF membrane (Trans-Blot Turbo, 0.2 µm, Bio-Rad) by a transfer system (Trans-Blot Turbo Transfer System, Bio-Rad) and blocked in 5% TEN-T milk for 1 h at RT prior to incubation with a primary antibody (overnight, 4 °C). Membranes were washed 3 × 20 min in TEN-T and incubated with the

secondary HRP-coupled antibody (5% milk/TEN-T, 1 h, RT). After a second round of 3 × 20 min washes, the signal was detected with ECL select, Lumino and Peroxide solutions (Amersham), and the Amersham Imager 600. Antibodies used are either commercial (95kd nuclear pore protein by Mab414; Abcam, ab24609) used 1:1000, Rabbit polyclonal antibody against hACT1 (aa 346–375; My BioSource, BSS9231831) used 1:1000, rat anti-Tubulin monoclonal antibody (Abcam, ab6161, YOL 1/34) used 1:10000 dilution, anti-IAA17 (gift of M. Kanemaki, NIG, Japan), used 1:1000, or anti-H2A-S129-phospho antibody (rabbit polyclonal), 1:1000 dilution[95] validated in cited reference. Also where indicated mouse anti-actin (monoclonal); (Clone C4, MAB1501, Millipore), was used at 1:4000 dilution. Full blots of those shown are in Source data.

### Phosphoproteomic analysis

The phosphoproteomic analysis to identify differentially phosphorylated proteins under conditions of YCS was performed as described in the text in triplicate. Mass spectroscopy was performed using a label-free method, and significance of change and GO-term designation is in ref. 57,58. Raw data and protein identification are uploaded as Supplementary Data 2.

### Detection of nuclear localization of Dam fusions

Constructs were based on yeast codon-optimized bacterial Dam, Las17-Dam, and Orc2-Dam fusions and were expressed from single copy plasmid from a *GAL1* promoter without induction on galactose (p415-GAL1 backbone[96], Supplementary Table 3). The plasmid transformants in GA-1981 wild-type cells were exponentially grown in synthetic medium minus leucine with 3% glycerol, 2% lactic acid, and 0.05% glucose and were incubated with or without 300 µg/ml Zeocin for 1 h at 30 °C. DNA was isolated with the MasterPure™ Yeast DNA Purification Kit (Lucigen). Genomic DNA was digested with *DpnI* (cleaves at G$^m$ATC) or *Sau3AI* (cleaves GATC, methylation aspecific; both New England Biolabs) overnight at 37 °C, subjected to 1% agarose gel electrophoresis in 0.5× TBE and was stained with SYBR safe (Thermo Fisher Scientific). Visualization and quantitation were performed with the Typhoon FLA 9500 scanner (GEHealthcare LifeSciences). For Southern blotting, the gel was transferred to the positively charged nylon membrane (Roche), and the membrane was probed with DIG-labeled rDNA (RFB region), followed by an anti-DIG-alkaline phosphatase-conjugated antibody (Roche). The signal was developed with CTD star substrate (Roche) and visualized as described. A diagram of DamID is derived from ref. 72, and the original procedure is as described[71].

### ELM search tool

The *mps2* mutant was designed based on a predicted actin-binding domain (aa83-98, found by the Eukaryotic Linear Motif Resource for Functional Sites in Proteins (ELM, http://elm.eu.org/search/) plus deletion of the protein upstream of this domain to eliminate potential additional N-terminal interactions with the cytoskeleton. Other deletions and mutations were made by standard yeast genetic approaches.

### Mammalian cell culture, viability, and immunofluorescence assays

Human dermal fibroblasts neonatal (HDFn) cells (Thermo Fisher Scientific) were cultured in Medium 106 supplemented with Low Serum Growth Supplement Kit (Thermo Fisher Scientific) at 37 °C in 5% CO$_2$. The number of cell death was measured by BIO-TC20 (BioRad) as Trypan blue positive cells, and the number of suspended cells was determined as Trypan blue negative cells released from the culture dish based on 60,000 total cells counter per condition. For phospho-Histone H2A.X detection, HDFn cells were seeded on the cover slide (Matsunami) and fixed with 4% paraformaldehyde (PFA) in PBS for 15 min at rt. Cells were washed once with PBS, treated with 0.5%

TritonX-100-PBS for 10 min, washed with PBS, and blocked with 2% BSA-PBS for 10 min. They were then incubated with 1:1000 diluted anti-phospho-Histone H2A.X (Ser139) antibody (JBW301 Merck Millipore) in PBS at 37 °C for 1 h. Cells were washed three times with PBS and incubated with 1:1000 diluted Alexa Fluor® 488 (Thermo Fisher Scientific) and 2 µM SiR-actin (Cytoskeleton Inc.) in PBS at 37 °C for 1 h. After three washes in PBS, cells were counterstained with 10 µg/ml 4',6-Diamidino-2-phenylindole, dihydrochloride (DAPI) in PBS for 2 min. Cells were washed once with PBS and then mounted with VECTA-SHIELD Mounting Medium (Funakoshi). Immunofluorescence images were acquired by Olympus IX83 with Olympus DSU Spinning Disk confocal using cell Sens software (Olympus).

The combinatorial compound assay using HCT116 cells was performed as described by Lehar et al.[53,97] and data is compared to a model for synergy calculations. Experimental conditions are described in Fig. 2e legend. The test assay data panel shows the % growth inhibition for the given concentrations of inhibitor. For the synergy over model panel, each measurement from the left panel was compared to expected values calculated from single agent dose-response curves to demonstrate synergy using the Loewe dose additivity (+ve values) or antagonism (−ve values)[53,97].

### Comet assay

Alkaline comet assay was performed with Comet Assay kit according to its instruction method (TREVIGEN, 4250-050-K) on HDFn cells cultivated as above. DNA was stained with SYBR Gold Staining solution (Thermo Fisher Scientific), and the images were acquired by Olympus IX83. Tail moment length was analyzed by CaspLab software[98].

### Yeast fluorescence microscopy

For microscopy experiments yeast was grown overnight in synthetic complete media containing 2% glucose. Then, the cultures were diluted to 1:40 in 5 ml and grown to OD$_{600}$ ~ 0.6. Cells were fixed in 4% PFA for 5 min, washed three times in PBS, and resuspended in a final volume of 500 µl PBS. For all imaging regimes, cells were fixed to a thin, SIM grade, Zeiss 1.5 glass coverslip using Concanavalin A.

All images were acquired on Olympus IX70 widefield microscope using a Prior Scientific Lumen 200 Pro illumination system. Photons were captured on an Andor Zyla 4.2 Plus sCMOS camera. Using a UPlan S APO ×100/1.4 oil objective, 40–60 Optical sections (50–200 ms exposure) of 0.2 nm were taken to section the whole plain of the yeast. A Semrock Penta-edge 4DB filter cube (408/504/581/667/762) was used with the following Semrock Brightline excitation and emission filter sets: DAPI 387/11, 440/40; GFP 485/20, 525/30, RFP (used for Mitotracker/Rh-phalloidin) 556/20 AFH, 607/36.

To stain mitochondria, Mitotracker (Invitrogen) CMXRos (5 nm) was added for 1 h prior to fixation. Actin filaments were stained by adding 10 µl of Rh-phalloidin (6.6 µm) to 100 µl of PFA fixed (4% PFA for 5 min) cells and left overnight. After washing in PBS, the cells were imaged using super-resolution microscopy as above. DNA content within cells was stained by adding 100 ng/ml DAPI solution to PFA fixed cells for 1 h followed by washing 3× in PBS. Images were deconvolved using Huygens Professional and channel-aligned using a custom Fiji (ImageJ) plugin. Images presented in the figures are maximum intensity Z-projections that have been scaled 3-fold using bilinear interpolation.

### Quantification of nuclear Las17-GFP

Cells expressing GFP-tagged Las17 (a C-terminal fusion which complements growth and Zeo resistance, see Supplementary Fig. 2f) were treated as indicated and fixed with 4% PFA at 4 °C overnight for microscopy. Cell nuclei were stained with DAPI (blue), and image stacks were acquired in both channels (10 planes, 200 nm distance). Nuclei that could be tracked across at least 5 planes were analyzed. Within a blue nuclear mask, the green average intensity was measured per plane and averaged across ≥5 planes to receive a single nucleus

mean intensity value. The number of measured nuclei is indicated in the figure legend. The level of significance was determined by the Wilcoxon rank sum test.

**Colocalisation of Las17/Pan1-GFP with MitoTracker**

Single plane images taken from stacks in both green (GFP) and red (MitoTracker) channels were thresholded automatically ("Moments dark") and the background was set to 0. Thresholded images were checked manually to remove out-of-focus images. Colocalization of the green with the red channel was determined per plane with the Cell profiler "MeasureColocalization" module. Plotted are single-plane values of Pearson's correlation coefficient. The number of cells measured was determined by counting nuclei on a maximum intensity projection of a stack in the blue channel (DAPI, nuclear staining) with the Fiji Cell Counter plugin.

**Actin quantitation**

Image acquisition of phalloidin stained and Cof1-RFP (Addgene #37102) expressing yeast cells, or cells treated with 0.25 mM IAA to induce Las17 degradation, as well as with 50 μg/ml Zeocin ± 0.5 μM CMB was carried out as described above[76]. Similarly, for actin Western blots the procedure described above was used with Rabbit polyclonal antibody against hACT1 (aa 346–375; My BioSource, BSS9231831 or anti-Tubulin antibody (Abcam, ab6161).

Budded cells with F-actin filaments were scored in maximum intensity projections of the image stacks acquired in the red channel. Three independent operators determined the total number of cells, the number of budded cells, as well as the number of budded cells with filaments with the CellCounter plugin in ImageJ/Fiji on blinded imaging data (for cells counted see figure). To quantify Cof1-RFP within the whole cells, a CellProfiler pipeline was created. Cells were identified with enhanced DAPI staining, and within a DAPI mask, the mean cellular Cof1-RFP intensity was measured in average projections of the stacks acquired in the red (Fig. 6e).

**Statement on reproducibility.** Single CHEF gels are shown in their entirety, and all were quantified, representing between one to three repeats of each experiment. YCS assays are performed precisely as described above. When only two exact repeats of an experiment were performed, it is noted in the legend. CHEF gel quantitation is provided as an Excel file in Supplementary Data 2. Western blots and imaging experiments were also repeated at least 3 times. The optimal concentration of Zeocin is determined batch by batch as the specific activity of the commercial product varies: we titrate each commercial batch received in a standardized assay for W303 yeast survival, and small aliquots are frozen and defrosted only once. We note that Zeocin is also light-sensitive.

**Reporting summary**

Further information on research design is available in the Nature Portfolio Reporting Summary linked to this article.

## Data availability

All data generated during this study are included in the main or supplementary material. The quantitation of CHEF gels is quantified in Supplementary data 2. Stacks of yeast cells imaged and quantified are available upon request to the corresponding author. Source data are provided with this paper.

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

## Acknowledgements

We thank Masato Kanemaki for anti-IAA17 antibody, Sue Jasperson, David C. Amberg, Anne Spang, and Kathryn Ayscough for useful strains. We thank Michael M. Stadler for the helpful discussion and for the sequence and computational analysis that identified the *cap2* mutation. We thank the FMI imaging facility for constant support, and the Gasser laboratory for critical reading of the manuscript and extensive discussions over the years. This work was supported by the Human Frontiers Grant: Actin and Actin-related proteins—probing their nuclear functions to S.M.G. and M.H. C.B.G. was supported by an FP7 Marie-Curie Intra-European Fellowship. S.M.G. thanks the Swiss National Science Foundation for grant number 31003A_176286, which supported V.H., K.C., A.S., and K.S., and the Novartis Research Foundation for many years of support.

## Author contributions

V.H., C.B.G., and K.S. contributed equally to experimentation, evaluation of the results, and writing of the manuscript; C.V.D.T. and K.C. contributed to quantitation of the results; K.C., A.S., and S.Y. contributed to the imaging and quantitation of yeast and mammalian cells; B.K. and S.B.H. contributed the human synthetic sensitivity experiment in Fig. 2e, f, and B.B. contributed the proteomics analysis in Fig. 3. M.H. supervised and funded A.S., and, K.S. and S.M.G. contributed equally to the supervision, writing and finalization of the manuscript.

## Competing interests

The authors declare that no competing interests.
