## [Transparent Peer Review file · Nature Communications]

Loss of cytoplasmic actin filaments raises nuclear actin levels to drive INO80C-dependent chromosome fragmentation

Corresponding Author: Professor Susan Gasser

Version 0:

Reviewer comments:

Reviewer #1

(Remarks to the Author)

In this manuscript, Hurst et al. try to answer the question of how interference with actin balance brings about chromosome shattering in budding yeast (YCS) upon Zeocin treatment. In the accompanying manuscript (Shimada et al.), the same lab reports that base excision repair at closely spaced oxidative lesions on opposite strands is responsible for generating excessive DNA double-strand breaks. Here, they pursue the actin signaling pathway to explore the mechanism by which actin contributes. As described in the review of the accompanying manuscript, the question is novel and interesting, and the use of Zeocin to induce YCS conditions is justified, as it presents a model situation for a general and fundamental question. However, although this manuscript is full of data and the experiments were performed to a high technical standard, it fails to illuminate the underlying mechanism, and we do not learn much beyond what was published by the same lab previously (Shimada et al., *Mol. Cell* 2013), i.e. that a perturbation of actin dynamics is responsible for the effect. The lack of a convincing model stands in contrast to a large set of poorly connected data, many of which seem uninterpretable with respect to the question.

Major issues:

1. The relevance of Las17 phosphorylation (or phosphorylation of other factors) remains unclear. The authors describe a very systematic approach where they use proteomics to narrow down those factors that change their phosphorylation status specifically under YCS conditions; however, Fig. 2E then shows that most of the relevant factors are phosphorylated independently of Zeocin treatment; Las17 is even phosphorylated independently of YPK signaling. Moreover, the phosphosite mutant of Las17 does not exhibit the expected phenotype, suggesting that Las17 phosphorylation is irrelevant for YCS.
2. The relevance of Las17 localization is unresolved. The authors show that its localization to mitochondria does not seem to affect the YCS phenotype, but although they do not detect the protein in the nucleus, they compare its situation to mammalian WASP (which is recruited to DSB foci – Fig. 5).
3. The experiments shown here do not give evidence whether and how Las17 affects nuclear F-/G-actin balance, and the regulation of nuclear G-actin is not shown for endogenous actin under YCS conditions. The actin mutants used here do not resolve this issue.
4. The experiments in mammalian cells are not convincing. The additive effect is not nearly as dramatic as in yeast (Fig. 7C: 8.1% of gH2AX positive cells in Zeocin versus 12.3% in Zeocin+LatB) and strongly depends on the cell line. This does not appear to reflect a shattering of the whole genome as it is observed in yeast.
5. Quantification of the CHEF gels is problematic. Although they do show lane traces here (unlike in the accompanying manuscript), they are very selective with the conditions for which they show these traces.
6. I do not understand the complex model depicted in Fig. 8 or see how the data support it. It appears as if the authors tried to put all their observations together here, but what actin is actually doing does not become apparent.

Reviewer #2

(Remarks to the Author)

This paper by Hurst et al. examines the link between TORC2 and events that interfere with repair of Zeocin induced DSBs. This paper builds on past work by the Gasser lab showing that TORC2-mediated actin filament regulation is required for genomic stability following Zeocin or IR-induced DSBs (Mol. Cell, 2013). They report that maintenance of the actin cytoskeleton is required for maintenance of genomic integrity following Zeocin Zeocin induced DSBs. They present a series of systematic studies indicating that the activation of genome instability arises from reduced actin polymerization and propose that the actin cytoskeleton regulates the sensitivity of yeast cells to Zeocin by preventing the accumulation of a "toxic" level of nuclear actin.

This is a novel, systematic study focusing on the role of the actin cytoskeleton in resisting YCS. Coupled with the accompanying paper, it should merit publication in Nature Communications, after the concerns/comments given below are adequately addressed. A major concern throughout this work, and the accompanying paper, is the lack of quantitation of DNA SBs from the CHEF gels for different cell types (e.g., wt vs. mutant) or under different conditions. This is somewhat surprising, given that other aspects of these studies incorporate sophisticated, high level quantitation techniques. Specific concerns and comments are outlined below:

Specific Comments:

P. 5 (lines 111-112): What about in the presence of Zeocin (i.e. increases cell permeability)? Authors should discuss that possible increased permeability was ruled out in Shimada et al., 2013 (tor2-V2126G mutant) and accompanying manuscript with glycosylase and APE1 mutants.

P. 7 (lines 149-153): These types of comparisons invoke the need for quantitation of the amounts of DSBs in each case (not simply 'more' or 'less'). See comments in accompanying manuscript.

P. 9 (lines 206-210 and lines 213-215): Again, these differences require quantitation.

P. 17 (lines 421-422): Again, quantitation of the DSBs makes this data much stronger; gel scans give a 'qualitative' picture and are not a linear representation of the signals.

Minor Comments:

P. 2 (lines 36-37): Clarify "... not detected in the nucleus on Zeocin nor under YCS-conditions...."

P. 3 (line 57): Define F-actin, e.g. "...F-actin (filamentous)...."

P. 3 (lines 58-60): Rewrite sentence to clarify

P. 4 (line 76): "...cells have led to the proposal that...."

P. 5 (line 104): "...accumulation in the nucleus...."

P. 10 (line 248): No Fig. 3F; Do you mean 3S1C-F??

P. 16 (line 403): "..... (Figure 6A, YPD)." Clarify 'YPD' lanes better

P. 17 (line 410): Delete "a strongly"

P. 17 (line 414): Don't you mean "Figure 6C"?

P. 17 (line 424): used in the presence

P. 18 (line 451): Change to: "...to avoid the first treatment interfering with the second...."

P. 22 (line 543): "...involved in polymerase...."

P. 23 (line 573): "...in mammals are efficiently repaired by XRCC1-mediated Short-patch BER <-Cite reference

Reviewer #3

(Remarks to the Author)

This submission by Hurst et al. probes the connection between YCS and actin. The basic observation is that combination of actin depolymerization drugs with zeocin triggers YCS in a manner reminiscent of TORC2 inhibition. The authors identify Las17 (yeast WASP) and other proteins as being phosphorylated during YCS. Ultimately, they propose that Las17 regulates the cytoplasmic pool of G-actin and therefore the balance of actin between the nucleus and the cytoplasm. Upon loss of Las17, the authors argue that G-actin becomes available to accumulate into the nucleus, form toxic filaments that interfere with DNA repair. This an attractive hypothesis, however alternatives could have been tested. The manuscript suffers from some of the limitations described for the Shimada et al. submission. How physiologically relevant YCS is to normal physiology and DNA repair? The signal from CHEF gels is not quantitated, which is a problem given the inherent variability of the method. Finally, the conclusions are not always supported by the data presented and some important experiments are missing.

Specific points:

This submission explores both YCS and DSB repair and at times, both terms YCS and DNA repair are used interchangeably, which could be misleading. In fact, the authors show that YCS is independent of homology-directed repair

(HDR).

Figure 1B and lane 132-133: "all chromosomes remained intact". Chromosomes in V2126G strain are in a much better shape than in the TOR2 strain. However, they show low level of fragmentation.

Figure 1D, 6th lane is the same experimental condition as Figure 1E, 9th lane. However, the fragmentation profiles look remarkably different. There are still intact chromosomes in 1D, whereas all chromosomes are fragmented into small fragments in 1E. This variability should be addressed by running multiple biological replicates, followed by quantitation of the intensity of the bands and calculating the average.

Lanes 144-145, Figure 1D: LatA phenocopies TORC2 inhibition. The gel presented suggest that LatA has a milder phenotype that TORC2 inhibition and therefore, LatA does not phenocopy TORC2 inhibition. Lanes 149, Figure 1E: "although LatA was less efficient than CMB". In contrast to Figure 1D, in which LatA has a milder phenotype than CMB, in Figure 1E, the phenotype is reversed: LatA has a phenotype that is at least as severe as CMB, compare lanes 7 and 9. This demonstrates that solid conclusions cannot be drawn from the observation of single gels.

Phosphoproteomics screen. The rationale to remove some of the low zeocin alone and then to include high zeocin was not entirely clear.

Figure 2E. The phosphorylation changes of Las17, a major focus of this manuscript, clearly indicate that changes at S380 are identical in *ypk* mutants whereas zeocin is present or not. Therefore, these changes are due to *ypk* inhibition alone. Contrary to the claim made several times in the manuscript (lane 197, lane 249, lane 515), Las17 phosphorylation is not YCS specific as it is solely related to *ypk*. Therefore, it is unclear what the phosphoproteomics data bring to the study.

Consistency in the experimental protocols would facilitate side-by-side comparisons. Figure 3 for example: 3 different concentrations of zeocin are used and 2 TORC2 inhibitors.

Pan1-AID mutation appears to protect from chromosome fragmentation: compare lane 2 and lane 12 (3A): is this reproducible?

Figure 3A, last lane and Figure 3B, last lane. Depletion of either Pan1 or Las17 in presence of zeocin and TORC2 inhibition yield more fragmentation than zeocin and TORC2 inhibition alone, suggesting that these genes are not epistatic with TORC2/zeocin.

Lane 213-214 (and lane 228), Figure 3C. "degradation of Las17 was epistatic (with LatA treatment)". This statement, which is a critically important conclusion, is not fully supported by Figure 3C. Figure 3C shows that IAA/LatA/zeocin (3C, lane 12) yield shorter DNA fragments than LatA/zeocin, lane 6 or IAA/Zeocin, lane 10, arguing against epistasis. This emphasizes again the need for biological replicates and quantitation.

Figure 3B, last lane and Figure 3D, last lane. Inhibition of TORC2 by BHS yields greater fragmentation than inhibition by CMB in Las17-AID strain. This goes against the claim that CMB is a better inhibitor.

Figure 4A. It would be useful to show the actual viability following Zeocin alone, loss of Las17 alone and the combination in addition to the normalized data. This would show how sick the cells are.

Lane 284-285, Figure 4C and 4D: colocalization of cortical actin with Las17. The data would benefit from showing a correlation coefficient.

Lane 268, Figure 3S1F: "Las17-GFP.. was only mildly sensitive to zeocin". It seems that the fusion strain is much more sensitive than WT, possibly around 10-fold.

Las17/WASP plays a critical role in promoting ARP2/3-dependent actin branching. WASP contributes to this process in 2 ways. It binds to G-actin, which is added to the growing actin filaments and it also binds to ARP2/3 complex triggering critical changes in conformation that initiate actin polymerization. The authors tested the hypothesis that Las17's role was to regulate G-actin pool (see below). However, they did not address the consequence of down-regulating ARP2/3, which could have provided valuable information.

Lanes 403-404, Figure 6B. In 6B, it seems that the levels of expression of ACT1 and ACT1nes are different: ACT1nes being expressed at higher level, especially when adjusting to controls. The authors conclude that accumulation of a filament-forming actin in the nucleus was toxic. Because ACT1 expression yields some chromatin-bound actin, it would be important to test ACT1-S14C. It would also be important to show that nuclear actin filaments are formed under these conditions using phalloidin.

I am not sure how to interpret the Act1-111 data. Act1-111 is not thought to interfere with polymerization, yet it results in dissociation of actin cables. Because of its impact on filament stability, one would expect that expression of Act1-111nes would have a weaker phenotype than ACT1nes expression, assuming that nuclear actin filaments are toxic.

Other points:

Lane 50, reference 4 does not seem to deal with DNA repair.

Lane 72-75. This statement is not accurate and should be modified. The laboratory of Robert Grosse has developed detection of nuclear F-actin using nanobodies. Furthermore, Dyke Mullins who originally reported the connection between nuclear F-actin and DNA repair used phalloidin to detect nuclear filaments.

Lane 86-87. "Thus, the question whether nuclear actin or actin binding proteins play a role in the repair of DNA damage remains open (19, 22, 23)". This sentence should be modified as it applies to yeast: "Thus in yeast, the question..." and only reference 22 should be mentioned.

In addition, work from the Mekhail's laboratory shows that nuclear microtubules play a role in DNA repair in yeast. This should be mentioned.

Lane 281-282. Rephrase: the correlation coefficient does not show that the structures are mitochondria.

Typos:

Lane 58: remove "In".

Lane 104: "in" the nucleus.

Lane 194: Figure 2E?

Lane 364: Figure 5C.

Version 1:

Reviewer comments:

Reviewer #1

(Remarks to the Author)

The authors have made a number of revisions to their manuscript. Two points in particular have been addressed: first, the quantification of their CHEF gels strengthens the quantitative YCS measurements overall. Second, they show a link of the YCS phenomenon to the actin-containing chromatin remodeler INO80.

This latter finding specifically strengthens the model that the authors put forth in the accompanying manuscript (Shimada et al). In my review of that manuscript, I mentioned that inclusion of the INO80 data would significantly enhance the impact of that study. Evaluation of this present manuscript further confirms my view on the pair of manuscript: I would strongly recommend moving the INO80 data into the Shimada et al manuscript; the resulting study would be a very nice addition to the journal.

In contrast, the Hurst et al. manuscript does not meet my expectations of a well-rounded mechanistic study in this journal. My previous impression (an extensive collection of relatively loosely connected data that lacks a stringent logical flow) remains unchanged, despite the authors' detailed explanations in response to my concerns. I still do not understand the relevance of their phosphoproteomics analysis to their conclusions. The section on mammalian cells remains isolated and without much mechanistic insight other than that the mechanism likely differs from the yeast situation. I do not wish to argue that the data shown here are not valuable, but I do not think that the format in which they are presented lends itself to a publication in this journal.

Reviewer #2

(Remarks to the Author)

The revised manuscript by Hurst et al. have successfully addressed most of my concerns/comments. Unfortunately, there are still issues with the quantitation methods they have used. I have addressed these new concerns with my responses to the rebuttal letter below:

1) P. 5 (lines 111-112): What about in the presence of Zeocin (i.e. increases cell permeability)? Authors should discuss that possible increased permeability was ruled out in Shimada et al., 2013 (tor2-V2126G mutant) and accompanying manuscript with glycosylase and APE1 mutants.

• OK

2) Next 3 comments: P. 7 (lines 149-153): These types of comparisons invoke the need for quantitation of the amounts of DSBs in each case (not simply 'more' or 'less'). See comments in accompanying manuscript. Now provided. P. 9 (lines 206-210 and lines 213-215): Again, these differences require quantitation. Provided. P. 17 (lines 421-422): Again, quantitation of the DSBs makes this data much stronger; gel scans give a 'qualitative' picture and are not a linear representation of the signals. Provided.

• As detailed in the review of the accompanying manuscript, the B/A ratio is flawed by systematic error (as opposed to statistical error), and the SSB/(unit DNA) should be calculated from the ensemble average of the corrected gel scans.

• These numbers will provide an accurate account of the gel data and match the high quality of the rest of the manuscript.

3) All Minor Comments:

• OK

Version 2:

Reviewer comments:

Reviewer #2

(Remarks to the Author)

In the revised manuscript by Hurst et al., the authors have successfully addressed all my previous concerns/comments. There are still a few minor issues with the text (a few examples are pointed out below) and the authors are encouraged to proof-read the manuscript once more to correct these. After the authors 'clean up' these minor issues, I think this manuscript will be ready for publication in Nature Communications. I would like to add that the process of 'review-response' over the last few years with this work has been a 'tour de force' and is a tribute to the persistence for experimental excellence by this lab.

Minor corrections:

p.8, line 192: "...detected a synergistic effect"

p.8, line 200: Label abscissa in Figure 2C

p.9: Haven't cited Figure 2F?

p. 14, line 358: Delete 1st "We" to give "Therefore, we...."

p. 19, lines 473-475: Cite reference for this statement

Michael J. Smerdon, Ph.D.
Regents Professor Emeritus, Biochemistry & Biophysics
School of Molecular Biosciences
Washington State University
Pullman, WA 99164-7520

REPLIES TO REVIEWERS' COMMENTS

This entirely reworked ms responds to all reviewers' comments with the following key additions:

- new data showing that depletion of the actin-containing remodeler INO80C results in resistance to shattering, reinforcing data in Shimada et al (NCOMMS-20-19157A) which shows that polymerase processivity and chromatin accessibility positively correlate with YCS.
- all our CHEF gels are quantified, providing numbers for the degree of shattering under each condition. All experiments that have been reproduced.
- globular vs filamentous actin levels (or filamentous vs total) are quantified under Las17 ablation and YCS conditions. We find an enrichment of actin in the nucleus.
- the Las17-GFP Zeocin sensitivity to shattering is shown to be like wild-type.
- the phosphoproteomics data interpretation is explained better
- DamID shows that Las17 is not nuclear under damage conditions, highlighting how yeast Las17 is different from its mammalian homolog WASP. Las17 lacks the nuclear localization signal that enables nuclear import
- a simpler and more compelling model is presented, and all comments are answered below

Reviewer #1 (Remarks to the Author):

In this manuscript, Hurst et al. try to answer the question of how interference with actin balance brings about chromosome shattering in budding yeast (YCS) upon Zeocin treatment. In the accompanying manuscript (Shimada et al.), the same lab reports that base excision repair at closely spaced oxidative lesions on opposite strands is responsible for generating excessive DNA double-strand breaks. Here, they pursue the actin signaling pathway to explore the mechanism by which actin contributes. As described in the review of the accompanying manuscript, the question is novel and interesting, and the use of Zeocin to induce YCS conditions is justified, as it presents a model situation for a general and fundamental question. However, although this manuscript is full of data and the experiments were performed to a high technical standard, it fails to illuminate the underlying mechanism, and we do not learn much beyond what was published by the same lab previously (Shimada et al., Mol.Cell 2013), i.e. that a perturbation of actin dynamics is responsible for the effect. The lack of a convincing model stands in contrast to a large set of poorly connected data, many of which seem uninterpretable with respect to the question.

We hope we have resolved the problem of “disconnected data” with a clear line of logic. In the first submission we were eliminating a series of alternative hypotheses based on results in other systems, without providing a compelling alternative. This is perhaps why it lacked a red line. We have now remedied this by providing data that support a compelling mechanism – that of higher levels of nuclear actin influencing nucleosome remodeler activity. This fits well with our findings on how disrupted BER causes DSBs, in the accompanying ms (Shimada et al. in review, included).

Major issues:

1. The relevance of Las17 phosphorylation (or phosphorylation of other factors) remains unclear. The

authors describe a very systematic approach where they use proteomics to narrow down those factors that change their phosphorylation status specifically under YCS conditions; however, Fig. 2E then shows that most of the relevant factors are phosphorylated independently of Zeocin treatment; Las17 is even phosphorylated independently of YPK signaling. Moreover, the phosphosite mutant of Las17 does not exhibit the expected phenotype, suggesting that Las17 phosphorylation is irrelevant for YCS. We believe there is a misunderstanding of Figure 3 here: Las17 phosphorylation on S380 increases when YPK is shut off with or without Zeocin (and thus does respond to YPK signalling); it also responds, albeit more weakly, to high Zeocin levels. We do not know what is the upstream trigger in the latter case. Obviously Ypk1/2 are not directly mediating S380p. Note that we are only detecting one of multiple Las17 phosphoacceptor site changes (proteomics can fail to recover peptides for a variety of reasons). A paper from the Ayscough lab (Tyler et al. 2021), describes how phosphorylation of the WH2 domain of Las17 reduces G-actin affinity *in vitro* (S380 is not in the WH2 domain). A phospho-mimetic mutant in this domain (S554) consistently reduces F-actin levels at endocytic sites as well, and this is what we observe under YCS conditions. Our screen does not detect Las17-S554p, yet that does not exclude that it is altered. Moreover, as these authors write, “despite the S554A mutation enhancing actin binding affinity and actin nucleation *in vitro*, the *in vivo* impact was to markedly inhibit the endocytic process both by delayed recruitment of Arp2/3 and of the inhibition of the subsequent invagination stage.” It is likely that other modifications contribute to Las17 control. Finally, we note that neither phosphorylation event on Las17 is mediated directly by Ark1 or Prk1; these modify Pan1 and Sla1, which are then lost from the cortical /endocytotic complex (Smythe and Ayscough, EMBO Rep 2003). Given that Pan1 binds to and regulates Arp2/3, which catalyses cortical actin polymerization, its absence down-regulates Arp2/3 (which is also the result of Las17 inhibition). Determining which phosphosite does what, in the Ypk/Ark/Prk cascade was not our goal, and hopefully the revised text makes this clearer.

In fact, as we explain in our manuscript, we used our phosphorylation data simply to highlight the fact that many actin cytoskeleton regulators change phosphorylation during YCS, not to characterize the precise regulatory event. It led us to delete or degra tag 5 players in cortical actin regulation (6, including Cap2). Of these, Las17 degradation had the most profound effect, mimicking LatA and TORC2 inhibition. It was not our intention to define how the Torc2 kinase cascade regulates Las17 (it is clearly very complicated; based on SDS gel shift, Las17 seems to reduce its net phosphorylation under YCS conditions, yet obviously S380 is upregulated). As for the S380 residue – indeed, we converted it to alanine without any pronounced effects on YCS nor Zeocin sensitivity. Despite that, it is positioned “strategically” between two polyproline domains that bind G-actin (Urbanek et al., 2013). The truncation of Las17 upstream of these polyproline domains strongly increased Zeocin sensitivity (Suppl. Figure 2F). We did not pursue the truncation, as it eliminated a large part of the protein and may have dominant negative effects. Inducible degradation is a better approach for an “acute” phenomenon such as YCS. Consistent with the fact that negative regulation of Las17 by S554 phosphorylation releases cortical actin, we show that the loss of Lat17 decreases F-actin, increasing the G-actin complement in the nucleus (Figure 6). Knowing this, we can conclude that the phospho-proteomics screen was useful as it correctly identified proteins that act downstream of Ypk1/2 to regulate actin filament turnover during YCS, even if we do not know which modification triggers this.

2. The relevance of Las17 localization is unresolved. The authors show that its localization to mitochondria does not seem to affect the YCS phenotype, but although they do not detect the protein in the nucleus, they compare its situation to mammalian WASP (which is recruited to DSB foci, Fig. 5). Determining Las17 subcellular location was very important in order to rule out that Las17 acts directly in repair, as argued by those using mammalian systems. The burden of proof was on us to show that Las17 is not acting in the nucleus, and we believe we do this convincingly (Figure 5). Documenting the mitochondrial localization of Las17 may not be directly relevant to chromosome fragmentation: but mitochondrial-BER crosstalk is a very plausible route for altering BER efficiency (Apl1 has two forms of the same protein – one nuclear and one mitochondrial; and the mitochondrial genome is known to be more sensitive to oxidative damage than the nuclear one). What we show now is that the degradation of Las17 decreases filamentous actin without altering overall actin levels (Figure 6), as does the TOR inhibitor CMB. These, like LatA, predispose to YCS. In sum, our data shows that Las17 displacement from cortical actin patches does correlate with an increase in globular actin and higher levels of nuclear G-actin.

3. The experiments shown here do not give evidence whether and how Las17 affects nuclear F-/G-actin balance, and the regulation of nuclear G-actin is not shown for endogenous actin under YCS conditions. The actin mutants used here do not resolve this issue.

We now address this directly: we include data on F-actin abundance, and then use an F/G-actin binder to determine the amount of total actin in the cell, and use this ratio to argue for changes in G-actin levels upon degradation of Las17 and under YCS conditions (new Figure 6). Other papers have documented that the polyproline domain of Las17 binds G actin (Urbanek et al., 2013), and we show that a truncation that removes this domain renders cell sensitive to Zeocin.

Given that there are no means to specifically monitor nuclear G-actin in budding yeast without generating artefacts, we believe that we have everything possible to analyse the impact of Las17 on the F/G actin ratio (new Figure 6).

4. The experiments in mammalian cells are not convincing. The additive effect is not nearly as dramatic as in yeast (Fig. 7C: 8.1% of gH2AX positive cells in Zeocin versus 12.3% in Zeocin+LatB) and strongly depends on the cell line. This does not appear to reflect a shattering of the whole genome as it is observed in yeast.

Your interpretation that the response of mammalian cells is less “dramatic” than yeast is indeed correct, but we disagree that the data are “unconvincing”. They are very well controlled and reproducible across multiple assays. Why is the response “weaker” than in yeast? First, we use cells that are as “normal” as possible (primary fibroblasts) and indeed, a 1h exposure of these cells to Zeocin and actin depolymerization does not trigger the dramatic chromosome fragmentation that is observed in yeast. Nonetheless, we find through the Comet assay that strand breaks increase on YCS conditions, although many may be ss breaks. Finally, we present an extensive study of the synergy between a TORC1/TORC2 inhibitor and Zeocin (as well as Zeocin and cytochalasin D which destabilizes actin filaments). These are robust results, recapitulating in HCT116 cells, which carry MLH1 deficiency, the synergistic effect seen in yeast. We do find some differences between cancer cell lines and primary fibroblasts, as cancer cell lines almost all carry repair pathway mutations. Another reason for the “weaker” impact in mammalian cells may be their preference for Short-patch BER over Long-patch

BER. The dedicated SP-BER pathway is extremely efficient (using XRCC1-Pol β -Lig III; all missing in yeast). Finally, one can imagine that γ H2AX signals don't increase synergistically if the repair pathway affected does not trigger a checkpoint kinase response.

One may ask, if human cells do not respond as dramatically as yeast – why show it ? Well, whenever we report the yeast results, we are immediately asked about human cells. Moreover despite being less dramatic, these data are robust and point to a negative role for nuclear actin in BER (see also Hurst et al., 2021 doi: 10.1091/mbc.E20-10-0680). This is in contrast to many reports claiming that nuclear actin filaments (and WASP) promote repair. The differences in BER regulation between yeast and man are relevant in this context. By the way, there is a positive side to the relative lack of sensitivity of mammalian cells to actin depolymerization and Zeocin: this combined therapy could be used to kill yeast infections with limited side effects in man.

5. Quantification of the CHEF gels is problematic. Although they do show lane traces here (unlike in the accompanying manuscript), they are very selective with the conditions for which they show these traces.

We have implemented a robust method for CHEF gel quantitation. Below each lane we provide the value of signal from below 0.55Mb divided by the signal above 0.57 Mb (the size of intact Chromosome 5) extracted from a normalized strip of image data, which is performed on different exposures. This gives the most robust means to compare chromosome fractionation that we have found. Details are in Shimada et al (under review) and in Materials and methods. 12 of the 16 yeast chromosomes (intact) are above this cut off point.

6. I do not understand the complex model depicted in Fig. 8 or see how the data support it. It appears as if the authors tried to put all their observations together here, but what actin is actually doing does not become apparent.

We now present a much simpler model based on the impact of INO80C mutants and the *act1-111* mutant. We propose that the actin perturbation we have investigated (due to TORC2 inhibition, LatA, Las17 degradation, or depletion of F-actin branching factors such as Sla1) all act by decreasing cytoplasmic F-actin and driving an increase in nuclear actin. The fact that elevated nuclear actin mimics these inhibitors is reported in an accompanying paper (Shimada et al. under review). In the nucleus increased G-actin alters the activity of actin-containing remodelers, such as INO80C, which leads to increased nucleosome remodeling or movement. This likely increases both DNA polymerase processivity and accessibility to base-modifying enzymes like Apn1 (cf Shimada et al., for impact of these on YCS). In brief, the actin-containing subcomplex of INO80 is known to be involved in nucleosome dynamics, INO80 promotes fork processivity under related conditions (Shimada et al., 2008; Papamichos and Peterson, 2008; Vincent et al., 2008), and finally, we found that chromatin accessibility increases on YCS conditions (Shimada et al., in review).

Reviewer #2 (Remarks to the Author):

This paper by Hurst et al. examines the link between TORC2 and events that interfere with repair of Zeocin induced DSBs. This paper builds on past work by the Gasser lab showing that TORC2-mediated actin filament regulation is required for genomic stability following Zeocin or γ IR- induced

DSBs (Mol. Cell, 2013). They report that maintenance of the actin cytoskeleton is required for maintenance of genomic integrity following Zeocin-induced DSBs. They present a series of systematic studies indicating that the activation of genome instability arises from reduced actinpolymerization and propose that the actin cytoskeleton regulates the sensitivity of yeast cells to Zeocin by preventing the accumulation of a “toxic” level of nuclear actin.

This is a novel, systematic study focusing on the role of the actin cytoskeleton in resisting YCS. Coupled with the accompanying paper, it should merit publication in Nature Communications, after the concerns/comments given below are adequately addressed. A major concern throughout this work, and the accompanying paper, is the lack of quantitation of DNA SBs from the CHEF gels for different cell types (e.g., wt vs. mutant) or under different conditions. This is somewhat surprising, given that other aspects of these studies incorporate sophisticated, high level quantitation techniques. Specific concerns and comments are outlined below:

Thanks for your positive feedback and helpful comments.

Specific Comments:

P. 5 (lines 111-112): What about in the presence of Zeocin (i.e. increases cell permeability)? Authors should discuss that possible increased permeability was ruled out in Shimada et al., 2013 (tor2-V2126G mutant) and accompanying manuscript with glycosylase and APE1 mutants.

It is clear that permeability is not the key factor driving YCS as γ IR does trigger fragmentation in combination with actin perturbation by either CMB (TOR inhibitor) or LatA (actin polymerization inhibitor). Nonetheless, γ IR is significantly less efficient than Zeocin (Shimada et al 2013). This is probably because the base lesions incurred are randomly distributed and not paired, as they are with bleomycin-family reagents (see Povirk, 1996; Shimada et al., in review). We took care in the accompanying ms to show that there is no major increase in oxidative lesions under YCS conditions (which would be the case if permeability were increased); rather the cause of YCS is blocked repair. Mutations in various BER factors attenuate the effect (Shimada et al, in review) as do remodeler mutations here (Figure 8). Moreover, impaired endocytosis due to actin depolymerization should lead to less Zeocin being taken up (i.e. less fragmentation), yet we observe the opposite. In sum, all these reasons suggest that differences in permeability are unlikely to be the driving factor for YCS.

P. 7 (lines 149-153): These types of comparisons invoke the need for quantitation of the amounts of DSBs in each case (not simply ‘more’ or ‘less’). See comments in accompanying manuscript. Now provided.

P. 9 (lines 206-210 and lines 213-215): Again, these differences require quantitation. Provided.

P. 17 (lines 421-422): Again, quantitation of the DSBs makes this data much stronger; gel scans give a 'qualitative' picture and are not a linear representation of the signals. Provided.

Minor Comments:

P. 2 (lines 36-37): Clarify “not detected in the nucleus on Zeocin nor under YCS-conditions.” changed
P. 3 (line 57): Define F-actin, e.g. “...F-actin (filamentous)...” done
P. 3 (lines 58-60): Rewrite sentence to clarify done
P. 4 (line 76): “...cells have led to the proposal that...” done
P. 5 (line 104): “....accumulation in the nucleus....” done
P. 10 (line 248): No Fig. 3F; Do you mean 3S1C-F?? These have all changed and are checked
P. 16 (line 403): “(Figure 6A, YPD).” Clarify ‘YPD’ lanes YPD is rich media, but this figure was removed; thus the comment is no longer relevant
P. 17 (line 410): Delete “a strongly” done
P. 17 (line 414): Don’t you mean “Figure 6C”? Figures have changed, but yes, we did
P. 17 (line 424): used in the presence done
P. 18 (line 451): Change to: “...to avoid the first treatment interfering with the second....” done
P. 22 (line 543): “....involved in polymerase....” done
P. 23 (line 573): “...in mammals are efficiently repaired by XRCC1-mediated Short-patch BER
<-Cite reference reviews cited- -Sam Wilson and others.

Thanks for your careful reading. We think we have resolved all these issues (page numbers and lines have changed however, although in some cases the comment is no longer relevant).

Reviewer #3 (Remarks to the Author):

This submission by Hurst et al. probes the connection between YCS and actin. The basic observation is that combination of actin depolymerization drugs with zeocin triggers YCS in a manner reminiscent of TORC2 inhibition. The authors identify Las17 (yeast WASP) and other proteins as being phosphorylated during YCS. Ultimately, they propose that Las17 regulates the cytoplasmic pool of G-actin and therefore the balance of actin between the nucleus and the cytoplasm. Upon loss of Las 17, the authors argue that G-actin becomes available to accumulate into the nucleus, form toxic filaments that interfere with DNA repair. This an attractive hypothesis, however alternatives could have been tested. The manuscript suffers from some of the limitations described for the Shimada et al. submission. How physiologically relevant YCS is to normal physiology and DNA repair? The signal from CHEF gels is not quantitated, which is a problem given the inherent variability of the method. Finally, the conclusions are not always supported by the data presented and some important experiments are missing.

We are happy that you agree in general that the topic is interesting and we hope we’ve supplied all the important missing experiments. We are careful only to draw conclusions fully supported by the data and leave speculation for the model.

Specific points:

This submission explores both YCS and DSB repair and at times, both terms YCS and DNA repair are used interchangeably, which could be misleading. In fact, the authors show that YCS is independent of homology-directed repair (HDR).

We have been more careful in the text discriminating YCS from impaired BER although they are largely identical phenomena. BER enzymes are responsible for YCS (Shimada et al., in review).

Figure 1B and lane 132-133: “all chromosomes remained intact”. Chromosomes in V2126G strain are in a much better shape than in the TOR2 strain. However, they show low level of fragmentation.

We quantify these CHEF gels so that one has a better idea of relative chromosome integrity. Although these are complex gels, I think we can state with confidence that the V2126G strain is resistant to YCS, arguing that fragmentation mainly goes through TORC2, and not TORC1. See previous work showing that Rapamycin (which inhibits TORC1) does not generate YCS.

Figure 1D, 6th lane is the same experimental condition as Figure 1E, 9th lane. However, the fragmentation profiles look remarkably different. There are still intact chromosomes in 1D, whereas all chromosomes are fragmented into small fragments in 1E. This variability should be addressed by running multiple biological replicates, followed by quantitation of the intensity of the bands and calculating the average.

Yes, there is variability in the degree of chromosome fragmentation which is in part to Zeocin itself. All experiments have been standardized as much as possible as the reviewer recommends, and replicated, often many times. Zeocin is highly sensitive to copper ions, to pH, and it decays naturally upon exposure to light (therefore we freeze the drug in small aliquots that we defrost once). Moreover, we titrate each batch and use the appropriate amount to trigger a given amount of damage. This is one reason that one sees a variation in the concentrations used. Another source of variation is the yeast background in which mutants are made (this is why we always use an isogenic wildtype strain as a control for each experiment). In some cases we had to use the S288C yeast background rather than W303, and the former is less sensitive to YCS. Concentrations of reagents must be adapted – within reason - to accommodate these strain background differences. Just FYI: there are over 300 gene-relevant differences between W303 and S288C backgrounds so the only good control is to include an isogenic wildtype control for each gel. We are constrained to use Zeocin because it generates clustered or “paired lesions” which underlie the mechanism of YCS (Shimada et al., in review). Generally our quantitation method overcomes Zeocin variability as long as we compare within one experiment. Finally, we note that the CHEF gel method requires that DNA is released from intact cells that are treated in an agarose plug prior to electrophoresis, posing another potential source of variability. For all these reasons we compare quantitative results on a gel-by-gel basis and ensure that the results are reproducible.

Lanes 144-145, Figure 1D: LatA phenocopies TORC2 inhibition. The gel presented suggest that LatA has a milder phenotype than TORC2 inhibition and therefore, LatA does not phenocopy TORC2 inhibition. Lanes 149, Figure 1E: “although LatA was less efficient than CMB”. In contrast to Figure 1D, in which LatA has a milder phenotype than CMB, in Figure 1E, the phenotype is reversed: LatA has a phenotype that is at least as severe as CMB, compare lanes 7 and 9. This demonstrates that solid conclusions cannot be drawn from the observation of single gels.

We agree that using the word “phenocopy” with respect to LatA is an overstatement, as TORC2 inhibition acts on multiple pathways, not only on actin dynamics. We clarify now that actin depolymerization is key to YCS with multiple lines of evidence. Both CMB and LatA depolymerize cytoplasmic F-actin and both cause chromosome fragmentation, but they are not 100% epistatic – clearly each condition has its own set of side effects. Moreover we show that these – like Las17 degradation, correlate with elevated nuclear actin (Figure 6). Coupled with the data in our accompanying paper, which shows that expression of nuclear export mutated actin alleles have the same impact as these inhibitors, we are able to pinpoint nuclear actin as the culprit in the observed conversion of paired abasic sites to DSBs. What causes differences in efficiency between LatA, Las17 degradation and TORC2 inhibition, is unclear.

Phosphoproteomics screen. The rationale to remove some of the low zeocin alone and then to include high zeocin was not entirely clear.

We clarify this in the figure legend. We thought there might be changes that are unique to YCS (low Zeocin combined with TORC2 pathway inhibition) and we did not want to be misled by changes triggered exclusively by DNA damage, whether low or high Zeocin levels. While we cannot exclude that the modifications contribute to damage and repair, they will not shed light on the Ypk1/2 regulated pathway in YCS.

Figure 2E. The phosphorylation changes of Las17, a major focus of this manuscript, clearly indicate that changes at S380 are identical in ypk mutants whereas zeocin is present or not. Therefore, these changes are due to ypk inhibition alone. Contrary to the claim made several times in the manuscript (lane 197, lane 249, lane 515)), Las17 phosphorylation is not YCS specific as it is solely related to ypk. Therefore, it is unclear what the phosphoproteomics data bring to the study.

We now clarify in the text that the phosphoproteomics confirm that actin dynamics are a major Ypk regulated pathway. We agree that the Las17 phosphorylation site may be irrelevant to chromosome fragmentation, but Las17 itself, and its ability to bind and sequester G-actin, *is* relevant, because Las17 regulates actin polymerization and is a G-actin chaperone. As described in our reply to Rev 1, its loss elevates G actin levels (Figure 6).

Consistency in the experimental protocols would facilitate side-by-side comparisons. Figure 3 for example: 3 different concentrations of zeocin are used and 2 TORC2 inhibitors.

The reason for changes in the TORC2 inhibitor is the higher efficiency of the newer chemical (see Shimada et al., Supplemental fig. 1). The reason we use slightly different Zeocin concentrations is because there is strong batch to batch variability in the purchased chemical. We titrate each batch in a standardized assay for W303 yeast survival, and then freeze small aliquots and defrost them only once. Different experiments are done with different batches (note that this study spanned nearly 10 years). If we use two concentrations in one experiment, it is in order to capture a range of Zeocin sensitivity. See Shimada et al., in review, for Zeocin titrations and drug titrations.

Pan1-AID mutation appears to protect from chromosome fragmentation: compare lane 2 and lane 12 (3A): is this reproducible?

From the newly quantified gels we do not see that Pan1-AID per se protects from fragmentation. Given its role in actin turnover and Arp2/3 control, it is expected that its loss might at least weakly mimic Las17 loss, which is what we observe.

Figure 3A, last lane and Figure 3B, last lane. Depletion of either Pan1 or Las17 in presence of zeocin and TORC2 inhibition yield more fragmentation than zeocin and TORC2 inhibition alone, suggesting that these genes are not epistatic with TORC2/zeocin.

We agree and this is mentioned now. Pan1 ablation is more additive with TORC2 inhibition than Las17. Note that the depletion through IAA takes longer than the usual treatment, so even though internal controls are performed, the temporal conditions for depletion experiments and pure inhibitor experiments are slightly different. Internal comparisons within the same experiment are the most reliable.

Lane 213-214 (and lane 228), Figure 3C. “degradation of Las17 was epistatic (with LatA treatment)”. This statement, which is a critically important conclusion, is not fully supported by

Figure 3C. Figure 3C shows that IAA/LatA/zeocin (3C, lane 12) yield shorter DNA fragments than LatA/zeocin, lane 6 or IAA/Zeocin, lane 10, arguing against epistasis. This emphasizes again the need for biological replicates and quantitation.

We agree that epistasis is difficult to call without robust quantitation, but we do not agree that it is “crucial” for our argument, especially given new evidence showing that nuclear actin is the culprit for altered BER, and that it is triggered both by Las17 degradation and TORC2 inhibition (Shimada et al., in review and Figures, 6,7). We minimize the use of epistasis arguments in the revised ms, although in some cases it still helps define a pathway. Note that all gels are now quantified (find values in Supplementary Data 5); all experiments are repeated multiple times.

Figure 3B, last lane and Figure 3D, last lane. Inhibition of TORC2 by BHS yields greater fragmentation than inhibition by CMB in Las17-AID strain. This goes against the claim that CMB is a better inhibitor.

In the accompanying ms we show in supplemental data that at given concentration CMB is about 10 times more efficient than BHS. However, some experiments were performed with BHS because we did not have CMB at the time.

Figure 4A. It would be useful to show the actual viability following Zeocin alone, loss of Las17 alone and the combination in addition to the normalized data. This would show how sick the cells are.

We did not retain Figure 4A (quantified colony forming assay) given the new focus of the paper, but include a similar drop assay for colony formation below for the reviewer. We have demonstrated that wild-type yeast survives well on low level Zeocin in many previous publications (e.g. Shimada et al, 2013, see Supplementary figure 1a; Hauer et al., 2016; see also Shimada et al, in review-attached), and that a short exposure (60-90 min) to either Zeocin or Las17 degradation confers almost no lethality, whereas the combination does. This is published (Shimada et al. 2013; also Hauer et al. 2016), and is shown in both in Supplemental Figure 2F and in Figure 2 for reviewers (below). On the other hand, persistent depletion of Las17 is lethal, because yeast requires actin filament dynamics for endo and exocytosis and thus for cell growth. Short exposure to the combination of Las17-AID/IAA with Zeocin, affects colony survival, while Las17-AID/IAA alone, or Zeocin alone (at the same concentration) does not. We include the colony survival data below to illustrate this (Reviewer Figure 1 below).

Lane 284-285, Figure 4C and 4D: colocalization of cortical actin with Las17. The data would benefit from showing a correlation coefficient.

Actually we think this data is less relevant to the paper, and instead we quantify the shift of Las17 to the mitochondria upon actin depolymerization (Figure 5E). The localization of Las17 at cortical actin patches is very well established in the literature (e.g. Robertson AS et al. The WASP homologue Las17 activates the novel actin-regulatory activity of Ysc84 to promote endocytosis in yeast. *Mol Biol Cell.* 2009).

Lane 268, Figure 3S1F: “Las17-GFP.. was only mildly sensitive to zeocin”. It seems that the fusion strain is much more sensitive than WT, possibly around 10-fold.

We have extensively tested the Zeocin sensitivity of our Las17-GFP strain with a range of Zeocin from 10 to 100 µg/ml and the strain is not 10 fold more sensitive (see Reviewer fig. 2). There is roughly a 5 - fold increase in Zeocin sensitivity at 50 – 100 µg/ml in a drop assay (5 fold dilution

series). In a YCS assay we see no increase in chromosome fragmentation on Zeocin, nor hypersensitivity to TORC2 inhibition. Viability and YCS do not strictly correlate (YCS is irreversibly lethal but cells can die from prolonged growth on Zeocin for other reasons). This fact is illustrated for cells at different stages of the cell cycle in the accompanying paper as well (Shimada et al., in review).

Las17/WASP plays a critical role in promoting ARP2/3-dependent actin branching. WASP contributes to this process in 2 ways. It binds to G-actin, which is added to the growing actin filaments and it also binds to ARP2/3 complex triggering critical changes in conformation that initiate actin polymerization. The authors tested the hypothesis that Las17's role was to regulate G-actin pool (see below). However, they did not address the consequence of down-regulating ARP2/3, which could have provided valuable information.

Arp2/3 in yeast has a strong impact on mitochondria, as well as cortical actin patches, and it is regulated by Pan1. While we could have pursued Arp2/3 or Pan1, the loss of Pan1 was less efficient and more additive with actin depolymerization (LatA or TOR inhibition) than Las17. Thus we do not see how pursuing these through a degradation assay would help us resolve how actin affects nuclear BER. In addition, even though the strong Arp2/3 effect on mitochondrial might be interesting per se, we felt it was beyond the scope of this paper. The impact of Arp2/3 on cortical actin patches is expected to be similar to that of Las17.

Lanes 403-404, Figure 6B. In 6B, it seems that the levels of expression of ACT1 and ACT1nes are different: ACT1nes being expressed at higher level, especially when adjusting to controls. The authors conclude that accumulation of a filament-forming actin in the nucleus was toxic. Because ACT1 expression yields some chromatin-bound actin, it would be important to test ACT1-S14C. It would also be important to show that nuclear actin filaments are formed under these conditions using phalloidin.

In our revised model, we propose that increased nuclear G-actin enhances the activity of actin-containing remodelers, which alters the processivity of DNA polymerases repairing adjacent base lesions, as well as access to Apr1. Experiments using a nuclear export deficient actin show that actin accumulation in the nucleus itself triggers shattering upon exposure of cells to Zeocin (Shimada et al., in review). This effect is slightly modulated by mutations that favor/disfavor polymerization, but the effect is not all-or-none. We think it is more important whether or not actin can be incorporated into actin-containing remodelers. Both of these actin variants are.

I am not sure how to interpret the Act1-111 data. Act1-111 is not thought to interfere with polymerization, yet it results in dissociation of actin cables. Because of its impact on filament stability, one would expect that expression of Act1-111nes would have a weaker phenotype than ACT1nes expression, assuming that nuclear actin filaments are toxic.

We now prove by quantitation that *act1-111* indeed shows less filaments as monitored by phalloidin staining (Fig 7). We attribute the Zeocin-sensitivity (and fragmentation on Zeocin) to the fact that having fewer cables means more G-actin is present in that mutant. Our results show that the *act1-111* mutation renders cells sensitive to Zeocin, but unresponsive to increased actin depolymerization or CMB. We think *act1-111* simply bypasses one branch of G-/F-actin buffering as act1-111 protein does not bind Las17 (Urbanek et al., 2013). As for the role of nuclear F-actin: we do not see any direct role of actin filaments in repair. The *act1-111* mutations should not interfere in INO80 assembly (see Reviewer Figure 3) and we detect Arp5 pulldown on chromatin in the *act1-111*

mutant. However, we were not able to purify biochemical amounts of INO80C with mutant actin (in part because these strains grow poorly), thus testing how the actin mutations affect the activity of actin containing remodelers *in vitro* is not possible.

Other points:

Lane 50, reference 4 does not seem to deal with DNA repair. Deleted

Lane 72-75. This statement is not accurate and should be modified. The laboratory of Robert Grosse has developed detection of nuclear F-actin using nanobodies. Furthermore, Dyke Mullins who originally reported the connection between nuclear F-actin and DNA repair used phalloidin to detect nuclear filaments.

There is increasing evidence that very rapidly turning over actin filaments form in mammalian cells. We have changed this statement, but the danger of using LiveAct-NLS tool persists.

Lane 86-87. “Thus, the question whether nuclear actin or actin binding proteins play a role in the repair of DNA damage remains open (19, 22, 23)”. This sentence should be modified as it applies to yeast: “Thus in yeast, the question...” and only reference 22 should be mentioned.

Done

In addition, work from the Mekhail’s laboratory shows that nuclear microtubules play a role in DNA repair in yeast. This should be mentioned.

We could not reproduce these results and therefore prefer to omit the reference to MTs.

Lane 281-282. Rephrase: the correlation coefficient does not show that the structures are mitochondria. Corrected

Typos:

Lane 58: remove “In”. Corrected

Lane 104: “in” the nucleus. Corrected

Lane 194: Figure 2E? Corrected

Lane 364: Figure 5C. Corrected

Data for reviewers

Reviewer Figure 1 : survival assay on YCS conditions To determine whether Las17 was required for recovery and growth after the repair of oxidative damage, or if it was important for repair itself, we triggered the transient degradation of Las17-AID during acute exposure to DNA damage (+Zeocin). We then plated cells in the absence of IAA and Zeocin, to monitor colony outgrowth. Whereas Las17 degradation did not compromise viability in the absence of damage, its degradation during the exposure to Zeocin reduced survival by 100-fold, consistent with the model that the absence of Las17 triggers irreversible damage when cells are exposed to Zeocin. The normalization is only performed to determine “1” at point zero, to normalize for the same number of starting cells.

Reviewer Figure 2: Las17-GFP zeocin sensitivity The reviewer's observation on the Zeocin sensitivity of Las17-GFP seen in a drop assay was confirmed by further drop assays. However, this sensitivity is mild (5 fold after 3 days growth on >50 $\mu\text{g/ml}$ Zeo, see below) and does not allow conclusions as to chromosome fragmentation. To rule out an effect on YCS we subjected this strain and its control to pulsed field gel electrophoresis under conditions of chromosome shattering (Zeocin + Tor inhibition by CMB). The results show that the Las17-GFP strain behaves like wild-type with respect to chromosome fragmentation (no sensitivity at 200 $\mu\text{g/ml}$ (red arrows) and normal fragmentation in presence of BHS and Zeocin (black arrows). Therefore, we conclude that despite the sensitivity of this strain to Zeocin in a drop assay, there is no difference with respect to the phenomenon investigated in our paper (YCS).

Figure 2 legend: The upper panels show first a drop assay for viability of yeast expression Las17-GFP. Shown are 5-fold dilution series on the indicated amount of Zeocin. Strains are indicated at left. Plates were incubated 3 days at 30C. The lower panel shows a typical YCS assay, with CHEF gel analysis of chromosome integrity (the panels blocked out are simply to focus the reader's attention on the relevant lanes for comparison). The Las17-GFP strain is around 5 fold more sensitive to 50 $\mu\text{g/ml}$ Zeo by drop assay, but shows no compromised chromosome integrity after 90 min with 200 $\mu\text{g/ml}$ Zeocin. Moreover, on BHS and Zeo, it shows as much fragmentation as WT (black arrows).

Reviewer Figure 3: *act1-111* should be able to form the INO80C subcomplex (Arp5-Arp8-Act1-Ies4-Taf14)

An image mapping the three *act1-111* mutations D222A, E224A, E226A, to the crystal structure of the Arp8-Actin interface. The mutations are on an exposed surface and are not expected to interfere with Arp8 nor its association with the holoenzyme. Instead the three D/E to A mutations are on an exposed surface that could interfere with its binding to other chromatin proteins.

The *act1-111* mutant is hypersensitive to Zeocin and is ts (at 30°C). We are able to recover Arp5 with chromatin, however, in the *act1-111* mutant, by chromatin fractionation (spheroplasting, gentle lysis and recovery of chromatin as a pellet, see below).

Rebuttal for **REVIEWER COMMENTS**

Nature Communications manuscript NCOMMS-20-19089A

Hurst, Gerhold et al

Reviewer #1 (Remarks to the Author):

The authors have made a number of revisions to their manuscript. Two points in particular have been addressed: first, the quantification of their CHEF gels strengthens the quantitative YCS measurements overall. Second, they show a link of the YCS phenomenon to the actin-containing chromatin remodeler INO80.

This latter finding specifically strengthens the model that the authors put forth in the accompanying manuscript (Shimada et al). In my review of that manuscript, I mentioned that inclusion of the INO80 data would significantly enhance the impact of that study. Evaluation of this present manuscript further confirms my view on the pair of manuscript: I would strongly recommend moving the INO80 data into the Shimada et al manuscript; the resulting study would be a very nice addition to the journal.

In contrast, the Hurst et al. manuscript does not meet my expectations of a well-rounded mechanistic study in this journal. My previous impression (an extensive collection of relatively loosely connected data that lacks a stringent logical flow) remains unchanged, despite the authors' detailed explanations in response to my concerns. I still do not understand the relevance of their phosphoproteomics analysis to their conclusions. The section on mammalian cells remains isolated and without much mechanistic insight other than that the mechanism likely differs from the yeast situation. I do not wish to argue that the data shown here are not valuable, but I do not think that the format in which they are presented lends itself to a publication in this journal.

The rigorously documented take-home from the Hurst et al manuscript is that there is a change in G-/F-actin balance upon TORC2 inhibition (and/or Las17 ablation), and that this interferes with oxidized base repair through nucleosome remodelers. The reason this is very important to the field is because the following papers (many of which were in *Nature* journals) claim that filamentous actin in the nucleus has either a positive role in DSB repair, or acts at stalled replication forks. We show that nuclear actin may increase remodeler activity and that it is deleterious to repair. These are the papers relevant to the question of actin in repair.

Zagelbaum et al., 2023 *Nat Struct Mol Biol.* 2023 30(1):99-106. doi: 10.1038/s41594-022-00893-6

Palumbieri et al. 2023 *Nat Commun* 2023 1:1 p 7819 DOI: 10.1038/s41467-023-43183-5

Caridi et al., *Nature* 2018 559 :7712 p 54-60 DOI: 10.1038/s41586-018-0242-8

Schrank et al., *Nature* 2018 559: 7712 p 61-66 DOI: 10.1038/s41586-018-0237-5

Nieminuszczy et al., 2023 *Nucleic Acids Res* 51:12 p 6337-6354 DOI: 10.1093/nar/gkad369

Han et al., *Nature Commun* 2022 13; 3743; DOI: 10.1038/s41467-022-31415-z

Along with papers on the positioning of DSB (or chromatin context) being important for repair such as Chen et al. *Nat Cell Biol* 2023, 25:1384

Schep et al *Molecular Cell* 2021 81(10), 2216–2230.e10. doi.org/10.1016/j.molcel.2021.03.032

Mitrensi et al., *Molecular Cell* 2022, 82:2132-2147 e2136.

see also : Belin et al., *E LIFE* 2015; Lamm et al., *NCB* 2020

The first 6 papers argue that actin and its chaperones (WASP and ARP2/3) generate nuclear actin filaments in the nucleus that are essential for damage clustering or processing (or in Caridi et al., dynamics), and that these guide DSB repair or replication fork stability. Later papers stress the

pathology of nuclear actin and 3D subnuclear clustering of damage. Our paper shows that in yeast WASP deletion generates DSB from ss lesions, but not due to the absence of nuclear F-actin nucleation by WASP and ARP2/3 at damage, but by upregulating actin-containing nucleosome remodelers due to enhanced accumulation of nuclear G-actin. This is an important alternative explanation of many of the data presented in these papers, and justifies a highly visible publication of Hurst et al.

The importance of the phosphoproteomics screen is that it identifies the effectors of the Ypk1/Ypk2 inhibition, triggered by TORC2 inhibitors. The most significant GO term for proteins with YCS-specific phosphorylation were linked to control of the cytoplasmic actin network and a specific set of complexes at the plasma membrane. Whereas other papers have implicated nuclear F-actin filament formation in DSB repair or replication fork stability, we show that the ablation of an actin chaperone does alter G-/F-actin balance and leads to altered BER, independently of DSB repair pathways and S-phase replication.

Reviewer #2 (Remarks to the Author):

The revised manuscript by Hurst et al. have successfully addressed most of my concerns/comments. Unfortunately, there are still issues with the quantitation methods they have used. I have addressed these new concerns with my responses to the rebuttal letter below:

1) P. 5 (lines 111-112): What about in the presence of Zeocin (i.e. increases cell permeability)? Authors should discuss that possible increased permeability was ruled out in Shimada et al., 2013 (tor2-V2126G mutant) and accompanying manuscript with glycosylase and APE1 mutants.

We now include this argument, and above all, we stress that abasic site generation is not enhanced by TORC2 inhibition. This would not be the case if TORC2 inhibition increased Zeocin uptake or activity.

2) Next 3 comments: P. 7 (lines 149-153): These types of comparisons invoke the need for quantitation of the amounts of DSBs in each case (not simply 'more' or 'less'). See comments in accompanying manuscript. Now provided, see changes throughout the ms's about comparative levels of DSBs.

P. 9 (lines 206-210 and lines 213-215): Again, these differences require quantitation. Now provided. P. 17 (lines 421-422): Again, quantitation of the DSBs makes this data much stronger; gel scans give a 'qualitative' picture and are not a linear representation of the signals. Now provided.

- As detailed in the review of the accompanying manuscript, the B/A ratio is flawed by systematic error (as opposed to statistical error), and the SSB/(unit DNA) should be calculated from the ensemble average of the corrected gel scans.

We believe this point arises from a misunderstanding of our gel system: we are not monitoring ss DNA nor ss fragments, only dsDNA and the average size of subchromosomal fragments. We refer to the comments about quantitation in Shimada et al (below), and we make it clear in the paper that we are only monitoring DSBs, not ss DNA accumulation nor SSB frequency (which as mentioned above may be up to 10 times the DSB frequency).

- These numbers will provide an accurate account of the gel data and match the high quality of the rest of the manuscript.

We hope that the quantitation we have provided on the frequency of DSBs below has clarified the issue.

3) All Minor Comments:
corrected

Response to **REVIEWER COMMENTS Nature Communications ms NCOMMS-20-19157A**

Shimada et al

Reviewer #1 (Remarks to the Author):

In the revised manuscript, Shimada et al. have successfully addressed many of my concerns/comments. Unfortunately, there are still issues with the new version, particularly with the *quantitation methods they have used*. In the following list, I have gone through the rebuttal comment by comment:

1)A major concern with this work is the lack of quantitation of SBs from the gels for different cell strains (e.g., wt vs. mutant) or under different conditions.

(p.24, 1st par.) As implied by the authors, the photo intensities are linear only over a small range and don't correlate with the actual pixel intensities of the scans. However, the Typhoon FLA 9500 response is linear over 5 orders of magnitude and Bio Rad chemiDoc XRS system has a dynamic range of 4 orders of magnitude. Therefore, one doesn't have to worry about linearity over the changing gel scan area due to changes in exposure.

We have clarified our quantitation methodology and we believe that the results we present are linear, robust and reliable. Because we were not sufficiently clear in our previous ms, there are several misunderstandings that led to the reviewer's comments. We have now provided extensive description of the gel conditions, the quantification methodology and have added figures that allow the reader to follow our arguments about the number of DSBs incurred, and the B/A quantitation. Relevant details are in Materials and Methods and the Supplemental Figure 2C-D. See also Figs 1 & 2 for reviewers at the end of this rebuttal.

To answer the points made above: first, we are not scanning photographs. We capture the fluorescence on the generally on the Typhoon FLA 9500 scanner with LPB (510LP) filter (GE Healthcare) or in early stages, with the Chem Doc XRS system (Bio-Rad). Both show a linearity of dsDNA detection over 10'000 fold dilution (see Figure 2 for reviewers below). The gel data were then directly transferred into Image J software.

For the quantitation, the rectangle which covers an entire lane was set as ROI, and DNA intensity was plotted. Background from a flanking region of the gel without sample was subtracted. We found no difference in monitoring intensity in Image J and the Typhoon scanner program, ImageQuant (see Figure 1 for reviewers). In the Image J program, a region of interest (ROI) spanning from above the largest chromosome band (2Mb) to ~20 kb (determined by size markers in the gel) was created and the signal intensity was measured using line plot profiling. The same ROI was applied for each lane in a CHEF gel image (note: in rare cases where the gels did not run straight, we adopted the ROI boxes appropriately. Note that because B/A is an "internal lane" ratio, this is still valid.

To measure B/A value, a division of the lane above and below 0.57 Mbp was created (above 0.57 Mbp includes Chr VIII/V bands through the largest Chr IV/XII; below extends from 0.56 Mbp (*below* Chr VIII/V) to about 20 kb, and includes most chromosome fragments and the small Chr I, III, VI, and IX). Whether or not there are ss gaps or nicks in the chromosome fragments is not relevant, nor is it scorable with nondenaturing gels. The intensity of ds DNA above (A) and below (B) 0.57 Mbp was measured. As long as the gels ran straight, the same ROI was shifted horizontally to measure all lanes in the gel. B over A was calculated and is indicated under each lane of each CHEF gel, representing the degree of

chromosome fragmentation. We are convinced that the B/A value comparison is robust and monitors irreparable DSBs which is what we aim to monitor.

We also modeled B/A ratios based on the mean fragment size detected by CHEF gels, and the values (see Supplemental Figure 2D) fit our measurements well. For B/A in the range of the experimental values (i.e., 0.2 – 13), the model predicts a mean fragment size of 136-624 kb (see plot in Supplemental Figure 2D). Note that the slope of the B/A curve around the mean fragment size of 100kb is very steep, so for mean fragment size of 100kb the predicted B/A is 50. We agree that mean fragment size is difficult to monitor if fragmentation is not complete, and therefore in the text we removed all references implying “x-fold” differences calculated by comparing B/A from different lanes, as requested.

Unfortunately, since the ImageJ program was used for quantitation of gel images, I assume screen shots of the data were analyzed. *This assumption is incorrect.* Because of the reason stated above, the analysis is more accurate when the file export information is directly analyzed. *This is what we did.* For the Typhoon data, this would be with the ImageQuant program. As stated below, the key is to obtain the number-average length (L_n).

We have the length of line data for all gels, but rigorous comparison showed that this was less robust than our method of integration of intensity within a ROI. We have systematically compared our method with others and are convinced this is the most reproducible method for fluorescent gel quantitation of “smears” of dsDNA created by random DSBs.

2)Indeed, this might reveal a critical ‘break frequency’ that is needed to generate YCS.

YCS arises from misregulation of BER, and we estimated the number of DSB incurred by 50-80 $\mu\text{g/ml}$ Zeocin (average amount used) to lie between 80-140 (for an average terminal ds fragment size of 100 kb \pm 20 we calculate 112 ± 22 DSB), in wild-type cells in the presence of TORC2 inhibitors. The number of ss lesions converted to DSBs depends on Zeocin concentration used (Shimada et al., 2013). We had already included modeling data that estimates the number of DSBs incurred to reduce the chromosomal complement to an average ds fragment size of a given mass (Shimada et al., Figures 1C and D). We now elaborate on this, discussing it in detail. We also extend the modeling for average fragment sizes from 70 to 700 kb in **Suppl Figure 2C** (see below). Note that the overall base oxidation frequency does not change with TORC2 inhibition (**Suppl Figure 2A**), but the *conversion* of ss lesions to DSBs does. Without TORC2 inhibition the levels of ss lesions induced are readily repaired.

(p.24, 2nd par.) The ratio chosen (B/A) is systematically flawed because the A region (>0.57 Mbp) contains long SSB fragments, and the amount changes with dose. Thus, this ratio would give a nonlinear dose-response (SSB/ μg vs [Zeocin]).

We are not monitoring ssDNA nor ss breaks, which would require a different staining reagent and probably alkaline gels. We use neutral *nondenaturing* gels and we cannot detect ssDNA as we use dyes that intercalate dsDNA. According to extensive literature, EthBr and SYBR safe dye (Invitrogen) are at least 10 fold more sensitive to ds than to ss DNA. Within a given experiment (with fixed levels of Zeocin) the number of **abasic sites** (resulting from glycosylase activity) is constant and this does not alter upon TORC2 inhibition (**Suppl Figure 2A**). In other words, we are exclusively monitoring the conversion of Zeocin induced base oxidation into DSB resulting from TORC2 inhibition, which occurs more readily after Zeocin or Bleomycin treatment, because these (closely related) reagents have a tendency to generate

“paired” or clustered oxidation events (Povirk 1996). Our goal was to score the relative rate of DSB generation from a fixed amount of Zeocin and determine how these arise. Povirk (1996) showed that bleomycin-like reagents (including Zeocin) produce ss nicks and ds breaks in a 10:1 ratio. Thus, it is reasonable to assume that the ss nicks are 10 times as abundant as the DSB numbers we score.

While we agree that DNA fragments carrying long-stretches of ssDNA might migrate aberrantly and more slowly, we also note that large ds chromosomal fragments that contain extensive ss stretches will be trapped in the well during PFGE (c.f. behavior during S phase, or upon HU arrest). Note that under our conditions (Zeocin ±CMB) we do not observe an increase in non-migrating high molecular weight DNA. This argues that chromosomes with ss nicks probably are migrating as intact chromosomes do, which is expected under neutral agarose (salt-containing) electrophoresis conditions.

The following text is now included to explain how we can model break formation from resulting average fragment size:

“Quantification of the mean number of DSB per chromosome was determined based on the assumption that DSBs are independently and uniformly distributed, i.e. they occur at the same frequency (λ) per unit length of DNA in all the chromosomes. Thus the number of breaks in chromosome i follows a Poisson distribution of parameter λS_i , where S_i is the size of chromosome i . In this model, the mean fragment size over all the chromosomes is $S_{tot}/(\lambda S_{tot}+16)$, where S_{tot} is the length of the genome. Hence the mean number of breaks per chromosome can be calculated as a function of the mean fragment size. Given an estimated mean fragment size in the YCS experiments (i.e. maximal intensity distribution curve) of 100 ± 20 kb, the corresponding mean number of breaks per chromosome are from Chr I to XVI are: 1.96; 6.92; 2.69; 13.04; 4.91; 2.30; 9.29; 4.79; 3.74; 6.35; 5.68; 9.18; 7.87; 6.68; 9.29; 8.07 (or 112 ± 22 total breaks; Fig. 1C-D). An extension of the graph in Figure 1C to an average fragment size of 700 kb is shown in **Supplementary Figure 2C**. A mean fragment size of 100-300 kb corresponds to a mean number of DSB genome-wide of about 25 (for 300kb average) to 110 (for 100kb average). Note that this is based exclusively on dsDNA detection (as fragments as well as full length chromosomes), on neutral gels that do not denature DNA. “

There are several papers that address the quantitation of these types of gels [e.g., Czaja et al. DNA Repair 9(9):976, 2010; Li et al. Sci. Reports 11(1):18393, 2021]. The background of undamaged chromosomes on CHEF gels can be subtracted to give a 'smear' which allows for determination of the ensemble average, the median length, & the number average length (L_n). It follows that $SSBs/(unit\ DNA) = 1/L_n (+Zeo) - 1/L_n (-Zeo)$.

Thank you for the suggestion. However, the Li et al paper is all alkaline gel assay and in the Czaja paper, all but one gel are alkaline denaturing gels, unlike our analysis.

3) the authors should use it to quantify the number of SSB vs. DSB from these gels for accurate comparisons of the different agents.

Thank you for the suggestion. However, as discussed above, using neutral CHEF conditions does not allow us to distinguish full length chromosomes from those with small internal stretches of ssDNA, or with ss nicks. In any case, what we monitor is *the relative rate of conversion of ss lesions to DSBs*, given a fixed level of base oxidation by a set Zeocin concentration in a given experiment. There is some variation in the amount of DNA loaded in each lane of a CHEF gel, as we are apply an agarose plug, which is why we decided to quantify using internal measurements of “small fragments <560 kb” vs “intact

chromosomes >570 kb". This yields a ratio specific for each lane, which gives a robust measure of DSB frequency, and the conversion of base oxidation to DSB.

It is important to note that the TORC2 inhibitor *does not increase the rate of oxidative lesions incurred* (abasic site frequency for a given amount of Zeocin is unchanged by TORC2 inhibition, **Supplemental Figure 2A**). Povirk (1996) estimated that bleomycin induces 10 times more ss nicks than DSB. We find no reason to challenge this. Interestingly, the smallest average size ds fragment (limit) we detect is in the *cdc9* mutant at nonpermissive temperature, which produces an average fragment size around 20kb, or roughly 600 DSBs per genome. We assume that loss of ligase 1 (*cdc9*) likely blocks all types of repair, (except that mediated by Lig4, which was not affected by TORC2 inhibition, Shimada et al 2013). More frequently we detect an average fragment size of 100 ± 20 kb, which represents a range of 112 ± 22 DSBs in wild-type cells. Obviously, most oxidized bases are not generating DSBs; yet it only takes a few irreparable DSBs to kill a cell (see survival plots after transient exposure to YCS conditions in Figure 4C,D)

4) authors should quantify the differences in overall strand breaks for different conditions; at least the percentage of breaks should be computed.

It is unclear to us what is meant here by "percentage of breaks". Percentage of DSB over ss lesions ? or over oxidized bases ? We believe this has been answered above (in wt cells it is 1:10; Povirk 1996).

5) P. 10 (line 228): Again, requires quantitation to make this statement.

We agree that one cannot convert B/A ratios into "absolute frequencies" of breaks, but one can calculate a B/A ratio graph that correlates with an average final product size (see panel B below). We removed all statements about "fold change" based on B/A ratios. We show in **Supplemental Figure 2D** the calculated B/A ratio for each value of λ (mean fragment size over all chromosomes ranging from 70 – 700 kb; Panel B) as follows:

A

B

Figure legend. **A**, For a given mean fragment size, we calculate the mean number of breaks for each chromosome and pile those numbers from chromosome 1 in the bottom to chromosome 16 in the top. Hence, the solid red line represents the mean number of breaks in the genome. **B**, B/A ratios from the model as a function of mean fragment size over all the chromosomes.

In the model, we assume that DSBs are independently and uniformly distributed, i.e. they occur at the same frequency (λ) per unit length of DNA in all the chromosomes (Cedervall and Källman 1994 – now

cited in the paper). Under these assumptions, the DSBs locations are described by a Poisson process with parameter λ . Thus, the number of breaks in chromosome i follows a Poisson distribution of parameter λS_i , where S_i is the size of chromosome i . In this model, the mean fragment size over all the chromosomes is $\frac{S_{tot}}{\lambda S_{tot} + 16}$, where S_{tot} is the length of the genome. Hence the mean number of breaks per chromosome can be calculated as a function of the mean fragment size (see panel A).

The theoretical relative intensity distribution, $I(x)$, of DNA fragment from randomly distributed DSBs is expressed as

$$I(x) = \lambda x e^{-\lambda x} (32 + S_{tot} \lambda - 16 \lambda x)$$

(see eqn. 2 in Cedervall and Källman 1994).

Therefore, the theoretical ratio B/A for e.g. A > 550 kb and B < 550 kb would be given by

$$\frac{\int_0^{550000} I(x) dx}{\int_{550000}^{\infty} I(x) dx}$$

Cedervall, B., and Källman, P. "Randomly Distributed DNA Double-Strand Breaks as Measured by Pulsed Field Gel Electrophoresis: A Series of Explanatory Calculations." *Radiation and Environmental Biophysics* 33, no. 1 (March 1994): 9–21.

For most of our B/A values (ranging from below 1 for intact chromosomes to values between 1 and 50) the mean fragment sizes range from 150-300 kb. Based on this graph, these B/A values estimates a from 80 to 20 DSBs genome-wide.

6) P. 12 (lines 260-261): Again, quantify.

Text has been appropriately correctly; all data are quantified.

7) P. 13 (lines 281-283): Could it be that DNA pol and/or RNA pol elongation blockage induce enhanced AP endo activity (i.e. G1 vs log-phase)?

We think this is unlikely because we see efficient conversion to DSBs in G1 (no DNA replication) and the addition of α -amanitin, which inhibits transcription does not mimic TORC2 inhibition (Shimada et al., 2013). We also observed that the acute Rpo21 (Rpb1, PolII catalytic subunit) degradation did not enhance DSB formation (not shown)

8) The more compact chromatin (on average) in G1-phase cells may partially 'release' BER enzymes from the 'hand-off' mechanism (e.g., Prasad et al., JBC, 2010) and APE activity may become more active.

We do not know of any data showing that chromatin is systematically more compact in G1 unless cells enter stationary phase, which is not the case here. But we do discuss that Apn1 activity varies with stages of the cell cycle (see Discussion) and show that chromatin accessibility changes with TORC2 inhibition.

It is confusing to tell if P/T panel goes with 7B or 7D. This should be clarified.

Corrected : the P/T panel is labeled independently.

9) Quantitation of the number of SB induced by Zeocin, \pm CMB, would address this possibility.

The number of abasic sites induced by a given amount of Zeocin is the same \pm CMB (Suppl Figure 2A).

Minor Comments

All OK Thank you

Minor Comments on new version:

p. 6 (lines 153-153): The statement "...nor is there any sign of DSB formation..." doesn't align with the data. Clarify. Removed

p. 17 (line 490): Delete "nuclei" actually the word "in" was missing. The phrase now reads "in nuclei" ...

p. 17 (line 496): ...subunit of [the] INO80 complex, which [is] an actin nucleosome..... Corrected

p. 24 (lines 720-722): Sentence is not clear and should be rewritten. Right – this is now corrected

Reviewer #2 (Remarks to the Author):

In their revised version, the authors have addressed many of the reviewers' original concerns and have strengthened their arguments substantially. This has resulted in a more coherent and convincing study.

The effect of Zeocin on DSB formation via the BER pathway is now very well documented.

The link to the actin system via the INO80 complex is again plausible, but still speculative because relevant functional data (e.g. effects of *ino80* mutants on YCS) are not provided. Such data appear to be in revision in an additional manuscript (Hurst et al) that the authors mention but do not show; including them in this manuscript would have helped the authors to make their point. Nevertheless, I am mostly happy with how the authors have responded to the reviewers' comments.

The inclusion of INO80 data in Shimada et al. would require first demonstrating that *in vivo* there is more G-actin in the nucleus. This demonstration is contained in 4 figures in Hurst et al. , which show that TORC2 inhibition and/or Las17 degradation drives higher nuclear G-actin levels. Without this demonstration, there is no proof that INO80C or other remodelers would be affected, and therefore the *ino80* mutant data fits better in the Hurst et al paper. We also note that the Hurst et al paper rules out other modes of action for nuclear actin, and that we test a range of nucleosome remodelers. These important experiments would not fit in a merged paper.

Rebuttal for **REVIEWER COMMENTS**

Nature Communications manuscript NCOMMS-20-19089A

Hurst, Gerhold et al

Reviewer #1 (Remarks to the Author):

The authors have made a number of revisions to their manuscript. Two points in particular have been addressed: first, the quantification of their CHEF gels strengthens the quantitative YCS measurements overall. Second, they show a link of the YCS phenomenon to the actin-containing chromatin remodeler INO80.

This latter finding specifically strengthens the model that the authors put forth in the accompanying manuscript (Shimada et al). In my review of that manuscript, I mentioned that inclusion of the INO80 data would significantly enhance the impact of that study. Evaluation of this present manuscript further confirms my view on the pair of manuscript: I would strongly recommend moving the INO80 data into the Shimada et al manuscript; the resulting study would be a very nice addition to the journal.

In contrast, the Hurst et al. manuscript does not meet my expectations of a well-rounded mechanistic study in this journal. My previous impression (an extensive collection of relatively loosely connected data that lacks a stringent logical flow) remains unchanged, despite the authors' detailed explanations in response to my concerns. I still do not understand the relevance of their phosphoproteomics analysis to their conclusions. The section on mammalian cells remains isolated and without much mechanistic insight other than that the mechanism likely differs from the yeast situation. I do not wish to argue that the data shown here are not valuable, but I do not think that the format in which they are presented lends itself to a publication in this journal.

The rigorously documented take-home from the Hurst et al manuscript is that there is a change in G-/F-actin balance upon TORC2 inhibition (and/or Las17 ablation), and that this interferes with oxidized base repair through nucleosome remodelers. The reason this is very important to the field is because the following papers (many of which were in *Nature* journals) claim that filamentous actin in the nucleus has either a positive role in DSB repair, or acts at stalled replication forks. We show that nuclear actin may increase remodeler activity and that it is deleterious to repair. These are the relevant papers relevant to the question of actin in repair.

Zagelbaum et al., 2023 *Nat Struct Mol Biol.* 2023 30(1):99-106. doi: 10.1038/s41594-022-00893-6

Palumbieri et al. 2023 *Nat Commun* 2023 1:1 p 7819 DOI: 10.1038/s41467-023-43183-5

Caridi et al., *Nature* 2018 559 :7712 p 54-60 DOI: 10.1038/s41586-018-0242-8

Schrank et al., *Nature* 2018 559: 7712 p 61-66 DOI: 10.1038/s41586-018-0237-5

Nieminuszczy et al., 2023 *Nucleic Acids Res* 51:12 p 6337-6354 DOI: 10.1093/nar/gkad369

Han et al., *Nature Commun* 2022 13; 3743; DOI: 10.1038/s41467-022-31415-z

Along with papers on the positioning of DSB (or chromatin context) being important for repair such as Chen et al. *Nat Cell Biol* 2023, 25:1384

Schep et al *Molecular Cell* 2021 81(10), 2216–2230.e10. doi.org/10.1016/j.molcel.2021.03.032

Mitrensi et al., *Molecular Cell* 2022, 82:2132-2147 e2136.

see also : Belin et al., *E LIFE* 2015; Lamm et al., *NCB* 2020

The first 6 papers argue that actin and its chaperones (WASP and ARP2/3) generate nuclear actin filaments in the nucleus that are essential for damage clustering or processing (or in Caridi et al., dynamics), and that these guide DSB repair or replication fork stability. Later papers stress the

pathology of nuclear actin and 3D subnuclear clustering of damage. Our paper shows that in yeast WASP deletion generates DSB from ss lesions, but not due to the absence of nuclear F-actin nucleation by WASP and ARP2/3 at damage, but by upregulating actin-containing nucleosome remodelers due to enhanced accumulation of nuclear G-actin. This is an important alternative explanation of many of the data presented in these papers, and justifies a highly visible publication of Hurst et al.

The importance of the phosphoproteomics screen is that it identifies the effectors of the Ypk1/Ypk2 inhibition, triggered by TORC2 inhibitors. The most significant GO term for proteins with YCS-specific phosphorylation were linked to control of the cytoplasmic actin network and a specific set of complexes at the plasma membrane. Whereas other papers have implicated nuclear F-actin filament formation in DSB repair or replication fork stability, we show that the ablation of an actin chaperone does alter G-/F-actin balance and leads to altered BER, independently of DSB repair pathways and S-phase replication.

Reviewer #2 (Remarks to the Author):

The revised manuscript by Hurst et al. have successfully addressed most of my concerns/comments. Unfortunately, there are still issues with the quantitation methods they have used. I have addressed these new concerns with my responses to the rebuttal letter below:

1) P. 5 (lines 111-112): What about in the presence of Zeocin (i.e. increases cell permeability)? Authors should discuss that possible increased permeability was ruled out in Shimada et al., 2013 (tor2-V2126G mutant) and accompanying manuscript with glycosylase and APE1 mutants.

We now include this argument, and above all, we stress that abasic site generation is not enhanced by TORC2 inhibition. This would not be the case if TORC2 inhibition increased Zeocin uptake or activity.

2) Next 3 comments: P. 7 (lines 149-153): These types of comparisons invoke the need for quantitation of the amounts of DSBs in each case (not simply 'more' or 'less'). See comments in accompanying manuscript. Now provided, see changes throughout the ms's about comparative levels of DSBs.

P. 9 (lines 206-210 and lines 213-215): Again, these differences require quantitation. Now provided. P. 17 (lines 421-422): Again, quantitation of the DSBs makes this data much stronger; gel scans give a 'qualitative' picture and are not a linear representation of the signals. Now provided.

- As detailed in the review of the accompanying manuscript, the B/A ratio is flawed by systematic error (as opposed to statistical error), and the SSB/(unit DNA) should be calculated from the ensemble average of the corrected gel scans.

We believe this point arises from a misunderstanding of our gel system: we are not monitoring ss DNA nor ss fragments, only dsDNA and the average size of subchromosomal fragments. We refer to the comments about quantitation in Shimada et al, and we make it clear in the paper that we are only monitoring DSBs, not ss DNA accumulation nor SSB frequency (which as mentioned above may be up to 10 times the DSB frequency).

- These numbers will provide an accurate account of the gel data and match the high quality of the rest of the manuscript.

We hope that the quantitation we have provided on the frequency of DSBs has clarified the issue.

3) All Minor Comments:
corrected

Figures for reviewers on gel quantitation:

Fig 1 Imagequant vs Image J

ImageQuant and ImageJ quantification B/A value comparison of SYBR safe stained CHEF gels.

Log2 scale

Fig 2 Linearity of SYBR safe dye and Typhoon scanning (our main tool). A two-fold dilution series of GeneRuler 1 kb DNA Ladder (Thermo Fisher Scientific) from 20 to 0.00975 ug of total ladder DNA were loaded on 1 % agarose gel and run in 1 x TAE buffer 100 V for 1h. The DNA was stained by SYBR Safe dye (Thermo Fisher Scientific) and scanned by Typhoon FLA9500. DNA bands were quantified by ImageQuant software and the intensity (arbitrary unit) was plotted over the DNA quantity (ng). We typically apply 95 - 240 ng total genomic DNA (0.8 - 2 E7 cells equivalent) in one lane for CHEF gel analysis, which corresponds to 2.1 - 30 ng of chromosomal DNA.

Final Reply to **REVIEWER COMMENTS**

Nature Communications manuscript NCOMMS-20-19089B

Hurst, Gerhold et al

Reviewer #1 (Remarks to the Author):

No further comments

Reviewer #2 (Remarks to the Author):

In the revised manuscript by Hurst et al., the authors have successfully addressed all my previous concerns/comments. There are still a few minor issues with the text (a few examples are pointed out below) and the authors are encouraged to proof-read the manuscript once more to correct these.

(done) After the authors 'clean up' these minor issues, I think this manuscript will be ready for publication in Nature Communications. I would like to add that the process of 'review-response' over the last few years with this work has been a 'tour de force' and is a tribute to the persistence for experimental excellence by this lab.

Minor corrections:

p.8, line 192: "...detected a synergistic effect" **corrected**

p.8, line 200: Label abscissa in Figure 2C **all abscissa are labelled on all graphs in figure 2 (a.u. is replaced by arb. u. for arbitrary units of fluorescence intensity) noted in legend**

p.9: Haven't cited Figure 2F? **it was cited together with Fig 2e (Fig 2e,f), now cited separately.**

p. 14, line 358: Delete 1st "We" to give "Therefore, we...." **corrected**

p. 19, lines 473-475: Cite reference for this statement **two references noting differences in W303 and S288c for INO80 dependence are cited.**